# A high-fat, high-sugar diet impairs maternal metabolism throughout pregnancy and lactation in mice

Stephanie E. O'Hara[1,2,3] (iD), Kelly M. Gembus[1,2,3] (iD), Georgia S. Clarke[2,3,4] (iD), Amanda J. Page[3,4] (iD), Kathryn L. Gatford[2,3,4] (iD) and Lisa M. Nicholas[1,2,3] (iD)

[1] *Adelaide Centre for Epigenetics, University of Adelaide, Adelaide, South Australia, Australia*
[2] *Robinson Research Institute, University of Adelaide, Adelaide, South Australia, Australia*
[3] *School of Biomedicine, University of Adelaide, Adelaide, South Australia, Australia*
[4] *Nutrition, Diabetes & Gut Health, Lifelong Health Theme, South Australian Health and Medical Research Institute, Adelaide, South Australia, Australia*

Handling Editors: Laura Bennet & Bruno Grassi

The peer review history is available in the Supporting Information section of this article (https://doi.org/10.1113/JP289985#support-information-section).

**Abstract figure legend** This study examined the impact of a high-fat, high-sugar (HFHS) diet consumed before mating and throughout pregnancy and lactation on maternal metabolic adaptation and early offspring outcomes in mice. Compared with controls, HFHS-fed dams showed greater adiposity, impaired glucose tolerance, altered fuel utilisation with higher fat and lower carbohydrate oxidation, and higher circulating insulin concentrations during a glucose challenge before pregnancy. Continued gestational exposure to a HFHS diet impaired normal adaptations to pregnancy, including preventing the shift toward carbohydrate utilisation, reducing fat deposition, and limiting pancreatic β-cell compensation for pregnancy-associated insulin resistance. Maternal glucose metabolism worsened across pregnancy and into lactation, with fasting hyperglycaemia, impaired glucose tolerance, altered insulin profiles and persistent changes in body composition evident at weaning. These maternal alterations were associated with fetal hyperglycaemia at gestational day 18 (GD18), which resolved by postnatal day 2 (PND2), and with increased adiposity in offspring at weaning.

The Journal of Physiology

**Abstract** Prenatal exposure to maternal overweight and elevated glucose increases risk of cardio-metabolic disease in offspring. Preclinical models such as the high-fat, high-sugar (HFHS) fed mouse allow mechanistic studies and testing of interventions, but it is first critical to understand the extent of exposures across early development. We therefore assessed the impacts of feeding a HFHS diet to C57Bl/6J mice for 11 weeks before mating and throughout pregnancy and lactation, on maternal weight, body composition, activity and energy expenditure, feeding behaviour, substrate utilisation and glucose metabolism. We also assessed the impacts of maternal diet on late gestation fetuses, neonates and early offspring growth. HFHS dams were fatter than controls with impaired glucose tolerance before mating and throughout pregnancy and lactation ($P < 0.001$). Dams also exhibited altered feeding behaviours, increased energy expenditure (light phase: $P < 0.001$, dark phase: $P < 0.001$) and a shift in fuel usage from carbohydrate to fat oxidation throughout pregnancy (lower respiratory exchange ratio: light phase: $P = 0.002$, dark phase: $P < 0.001$). Fetuses of HFHS dams were hyperglycaemic at gestational day 18 ($P = 0.031$). Altered patterns of offspring growth during lactation resulted in fatter pups at weaning. Consumption of a HFHS before and throughout pregnancy and lactation exposes offspring to changes in maternal metabolism *in utero* and throughout lactation. Since maternal impacts differ between studies, it is essential that these are characterised in each model to understand the critical factors that drive programming of offspring metabolism.

(Received 31 August 2025; accepted after revision 20 January 2026; first published online 3 April 2026)

**Corresponding authors** L. M. Nicholas and K. L. Gatford: Level 6, Adelaide Health and Medical Sciences (AHMS), Adelaide University, North Terrace, Adelaide, South Australia 5000, Australia.    Email: lisa.nicholas@adelaide.edu.au and kathy.gatford@adelaide.edu.au

**Key points**

- Consumption of an obesogenic, high-fat, high-sugar (HFHS) diet impairs glucose tolerance during pregnancy, but how this impacts metabolic adaptations during pregnancy and lactation remains unclear.
- In the present study, consumption of a HFHS diet in mice increased adiposity and impaired glucose tolerance before mating and throughout pregnancy and lactation.
- HFHS consumption impacted metabolic adaptations to pregnancy, including failure to shift from fat to carbohydrate oxidation, reduced fat deposition and lower insulin secretion.
- These alterations in maternal metabolism during pregnancy resulted in fetal hyperglycaemia and altered patterns of offspring neonatal growth, resulting in offspring that were fatter at weaning.
- These findings have implications for metabolic health of both mothers and their offspring.

## Introduction

The prevalence of cardiometabolic diseases such as obesity, type 2 diabetes and hypertension is increasing worldwide. It is well accepted that offspring that are exposed to an adverse intrauterine environment are at greater risk of longer-term health problems including obesity, type 2 diabetes and cardiovascular disease (Fernandez-Twinn et al., 2019; Petry & Hales, 2000). The increasing incidence of maternal overweight and obesity

**Stephanie E. O'Hara** is a PhD student in the School of Biomedicine at the University of Adelaide. She is interested in metabolism and maternal health, which are central to her research project investigating how maternal obesity and hyperglycaemia influence type 2 diabetes risk in offspring. Her work focuses on understanding the molecular mechanisms underlying metabolic disorders, with the goal of improving health outcomes for both mothers and children.

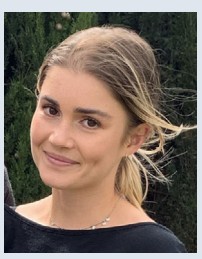

as well as hyperglycaemia in pregnancy (Kent et al., 2024; Shah et al., 2021) may therefore be contributing to increased prevalence of non-communicable diseases globally. In humans, *in utero* exposure to maternal obesity is associated with a 3.5-fold increased risk of type 2 diabetes in offspring (Lahti-Pulkkinen et al., 2019). Increasing maternal glucose concentrations during pregnancy are similarly associated with markers of type 2 diabetes risk in children including elevated fasting glucose, poorer insulin sensitivity and reduced insulin secretion (Scholtens et al., 2019).

Pre-clinical models are valuable tools to understand mechanisms mediating the impacts of these exposures to an adverse *in utero* environment on offspring health and to evaluate interventions. In many mouse models, this is achieved by maternal consumption of a diet high in both fat and sugar, which reflects a human obesogenic Western-style diet (Nicholas & Ozanne, 2019). To maximise the translational potential of these models, it is important to understand their impacts on maternal physiology and offspring development throughout the duration of exposure, often including lactation as well as pregnancy. Whilst it is well established that feeding female mice an obesogenic diet leads to impaired glucose tolerance during pregnancy (Fernandez-Twinn et al., 2017; Musial et al., 2017), most studies assess glucose metabolism at a single gestational time point, typically mid- or late pregnancy. Only a few have examined changes before and during pregnancy (Carter et al., 2015; Huypens et al., 2016; King et al., 2013; Sasson et al., 2015), and findings vary depending on diet composition, duration of pre-pregnancy feeding, and pre-mating adiposity (Nicholas & Ozanne, 2019). To date, no study has comprehensively assessed maternal glucose metabolism during lactation in a diet-induced obesity model, leaving a major gap in understanding the timing of, and persistence or resolution of, metabolic dysfunction postpartum.

The aim of this study was, therefore, to determine the impact of feeding mice an obesogenic high-fat, high-sugar (HFHS) diet on maternal weight and body composition, and glucose metabolism from before pregnancy through to the end of lactation in mice. We also determined the effect of the HFHS diet on feeding behaviour and energy metabolism throughout pregnancy. We further characterised the model by assessing its impact on fetal circulating glucose and insulin concentrations and size and neonatal growth.

## Methods

### Ethical approval

All experiments were approved by the South Australian Health and Medical Research Institute (SAHMRI) Animal Ethics Committee (Approval number: SAM-20-053). Studies were carried out in accordance with the Australian code for the care and use of animals for scientific purposes (National Health & Medical Research Council of Australia, 2013) and conform to the principles and regulations described by Grundy (2015).

### Experimental design

This study was performed in GLU-Venus mice on a C57Bl/6J background, which express the Venus fluorescent marker in $\alpha$-cells (Reimann et al., 2008). Together with autofluorescence of $\beta$-cells, this label enables sorting of isolated islet cells into these two populations for functional studies. Frequent backcrossing to C57Bl/6J mice means that the GLU-Venus mice are metabolically identical to this commonly used strain. In brief, 4-week-old female GLU-Venus mice were fed *ad libitum* either standard laboratory chow (Control group: Teklad Global 18% Protein Rodent Diet, digestible energy from protein 24%, fat 18% and carbohydrate 58%, 13 kJ/g; Envigo, Huntingdon, UK) or a high-fat, high-sugar diet (HFHS group: Specialty Feeds SF21-003, digestible energy from protein 17.6%, fat 58.4% and carbohydrate 24% [sucrose 175 g/kg], 23 kJ/g; Glen Forrest, Western Australia, Australia) for 11 weeks before mating. Feeding of experimental diets commenced at 4 weeks, approximately 1 week after weaning, when mice from the breeding colony are transferred to experimental holding areas. Mice were maintained on their respective diets throughout pregnancy and lactation (Clarke et al. 2026). Body weight and composition was determined using an EchoMRI Composition Analyzer (EchoMRI, Houston, TX, USA) in female mice prior to mating. Male breeders were fed standard laboratory chow except when housed with females for mating. Mice were housed in individually ventilated cages and maintained in a humidity-controlled room with a 12 h light–dark cycle, with free access to food and water. Female mice were housed in groups of three to five per cage prior to pregnancy.

Female mice were pair-housed with a GLU-Venus male mouse for no more than 7 days for mating. Female mice remained on their respective diets during this period. Pregnancy was confirmed by the presence of a vaginal plug, which was designated as gestational (GD) day 1. A vaginal plug was observed within the first 4 days of mating in 100% of chow-fed mice and 96% of HFHS-fed mice. Once mated, female mice were housed singly. Changes in body weight and composition during pregnancy (GD 1, 7, 14 and 18) were determined using an EchoMRI Composition Analyzer. At GD 18, a subset of pregnant dams was anaesthetised by isoflurane inhalation (5% in oxygen) and humanely killed

by decapitation. Fetuses and placentas were individually dissected and weighed. Placental efficiency was calculated as the ratio of fetal to placental weight. Fetuses were sexed by visual inspection of the anogenital distance, with males exhibiting a greater anogenital distance than females (Wolterink-Donselaar et al., 2009). Fetal body size was measured before fetuses were decapitated and trunk blood collected for later measurement of blood glucose and plasma insulin concentrations. When uncertain, sex of sampled fetuses was confirmed by dissection following post-mortem. Blood glucose was measured in individual fetuses from independent litters and insulin was measured in plasma pooled from same-sex fetuses within each litter.

The remaining dams gave birth naturally. At post-natal day (PND) 2, pups were sexed by visual inspection of the anogenital distance and perineal pigmentation, with males exhibiting a greater anogenital distance and a distinct pigmentation spot on the perineum, which is absent in females (Wolterink-Donselaar et al., 2009). Litters were standardised to five pups per dam, which remained with the dam for subsequent studies. Where additional pups were available in the litter, one pup per sex per litter was randomly selected for study at PND 2. When uncertain, sex of sampled fetuses was confirmed by dissection following post-mortem. Crown–rump length, abdominal transverse diameter, head length and biparietal diameter were measured using a digital calliper. Blood glucose concentration was measured in trunk blood collected from individual neonates following decapitation, using an Accu-Chek Performa glucometer (Roche, Basel, Switzerland). Blood from individual neonates was also collected into a heparinised capillary tube for plasma insulin measurement using the STELLUX Chemi Rodent Insulin ELISA (ALPCO, Salem, NH, USA). Maternal body weight and composition during lactation were determined 7, 14 and 20–22 days (weaning) after birth of pups using an EchoMRI Composition Analyzer. Because not all mice conceived and some dams were humanely killed for fetal studies, the groups studied pre-mating, throughout pregnancy and throughout lactation include different and partially overlapping subsets of female mice. Offspring body weight was measured at PND 2, 7, 14 and weaning and offspring body composition was determined at weaning using an EchoMRI Composition Analyzer.

## Metabolic monitoring

Two subsets of female mice were housed in metabolic cages (Promethion, Sable Systems International, Las Vegas, NV, USA) for 8 days prior to mating, consisting of a 4-day acclimatisation period followed by 4 days of pre-pregnancy measurements. Following this, female mice were pair-housed with a stud male in normal home cages for mating. They were returned to their individual metabolic cages on the morning when a vaginal plug was observed and remained in these cages until either GD 18 or PND 2. Caloric intake, feeding behaviour, activity and energy expenditure data were obtained from mice with data available to PND 2. Due to technical issues with gas analyses, fat and carbohydrate utilisation up to GD 18 were obtained from a subset of those housed in metabolic cages until PND 2 plus additional animals studied to GD 18, for which food intake data has been reported previously (Clarke et al., 2025). Food intake was defined as the reduction in food hopper weight, measured by high-precision sensors in real-time with 3 mg resolution. Meals were defined as a reduction in food hopper weight with a minimum food intake duration of 20 s. Meal size was defined as the reduction in food hopper weight during each meal. Meal duration was defined by the time spent interacting with the hopper during the meal (Ladyman et al., 2018). Data were transformed using the Promethion data software package, ExpeData v.1.9.14, using analytical macro 6, which analysed data in 12 h time periods corresponding to the light and dark periods of each day. Data are presented as means across the 4 days prior to mating for 'pre-pregnancy', GD 1 to GD 7 for 'early pregnancy', GD 8 to GD 13 for 'mid-pregnancy', and GD 14 to GD 18 or to the day before birth for 'late pregnancy'. Of a total of 17 mice, the majority (13 mice) gave birth on GD 20, two on GD 21 and one each on GD 19 and GD 22. Data points were excluded if they did not include a full 12 h of data for each photoperiod, for example, due to cage changes (Ladyman et al., 2018; Li et al., 2021).

Total carbohydrate and fat oxidation (g/h) were calculated from $\dot{V}_{CO_2}$ (L/h) to $\dot{V}_{O_2}$ (L/h) as described previously (Bhandarkar et al., 2021):

- Carbohydrate oxidation g/h $= 4.585 \times \dot{V}_{CO_2} - 3.23 \times \dot{V}_{O_2}$
- Fat oxidation g/h $= 1.69 \times \dot{V}_{O_2} - 1.69 \times \dot{V}_{CO_2}$

## Intraperitoneal glucose and insulin tolerance tests

An intraperitoneal glucose tolerance test (ipGTT) was performed 2 weeks before mating, at GD 16 and on the day of weaning. An intraperitoneal insulin tolerance test (ipITT) was performed 1 week prior to mating, at GD 18 and 2 days after weaning. Following a 6 h fast from 08.00 h, mice were given an intraperitoneal injection of either glucose (ipGTT, 2 g/kg) or insulin (ipITT, 0.5 U/kg, Actrapid, NovoNordisk, Bagsværd, Denmark). Blood glucose was measured using an Accu-Chek Performa blood glucose meter (Roche) at intervals for 90 (ipITT) or 120 min (ipGTT). During the ipGTT, tail vein blood was also collected into heparinised capillary tubes at 0, 15 and 30 min for plasma insulin analysis using the STELLUX Chemi Rodent Insulin ELISA (ALPCO). Areas under the curve for glucose and insulin responses

to glucose challenge were calculated relative to fasting concentrations of glucose and insulin, measured directly before the intraperitoneal glucose tolerance test. Area above the curve for the glucose response to insulin challenge was similarly calculated relative to fasting glucose measured directly before the intraperitoneal insulin tolerance test.

### Statistical analyses

Data are presented as the mean ± SD unless otherwise indicated, with $P < 0.05$ accepted as significant. Statistical analyses were performed in IBM SPPS Statistics 28 (IBM Corp., Armonk, NY, USA). For analyses involving GD 18 fetuses, placentas and PND 2 offspring, $n$ represents the number of mice from independent litters. The effect of HFHS feeding on pre-pregnancy body weight and composition was analysed using Student's unpaired two-tailed $t$ test. Changes in body weight and composition during pregnancy and lactation, blood glucose levels over time during an ipGTT and ipITT, and mean pup body weight between PND 2 and weaning were analysed using a two-way repeated measures ANOVA. Changes in plasma insulin over time during the ipGTT and metabolic cage data were analysed using a linear mixed model. Placental and fetal weight and fetal growth measures were also analysed using a linear mixed model with litter size included as a random effect. All other data were analysed using a two-way ANOVA. When a statistically significant interaction was present, a simple main effect analysis was performed.

### Results

### Mice fed a high-fat, high-sugar diet have increased adiposity prior to and throughout pregnancy and lactation

Not surprisingly, female mice fed a diet high in fat and sugar for 11 weeks were heavier and had increased fat mass relative to body weight prior to mating (Fig. 1*A* and *B*). This was accompanied by a reduction in relative lean mass (Fig. 1*C*). Pregnancy was associated with body weight gain independent of maternal diet, with HFHS mice weighing more than controls (Fig. 1*A*). Effects of diet on adiposity varied across pregnancy (Interaction $P < 0.001$). HFHS mice were fatter than controls throughout pregnancy. The adiposity of HFHS mice peaked on GD 7 and declined thereafter (Fig. 1*B*). In control mice, adiposity remained elevated from GD 7 through to GD 18 (Fig. 1*B*). HFHS mice also had lower relative lean mass during pregnancy, while pregnancy itself was associated with a decrease in relative lean mass on GD 7 and a further decrease on GD 14 (Fig. 1*C*). By late gestation, relative lean mass was comparable to the start of pregnancy. Changes in body weight during lactation differed between groups (Interaction $P = 0.005$, Fig. 1*A*). A week after giving birth, dams fed a HFHS diet were 6.2% lighter than controls; however, this difference in body weight between groups did not persist (Fig. 1*A*). Both control and HFHS dams gained weight between days 7 and 14 of lactation, which was followed by a period of weight loss between day 14 and weaning (Fig. 1*A*). Relative fat mass increased throughout lactation and was higher in HFHS mice than controls (Fig. 1*B*). Whilst relative lean mass did not change during lactation, it was lower in dams fed a HFHS diet compared to controls (Fig. 1*C*).

### Food intake and feeding behaviour change during pregnancy and are disrupted in mice fed a high-fat, high-sugar diet

Changes in light phase food and caloric intakes, meal size and meal number during pregnancy differed between groups (Interaction $P < 0.02$ for all, Fig. 2*A–D*). Despite lower food intake and smaller meal size during the light phase throughout the whole of pregnancy, caloric intake was lower in the HFHS compared to the control group only at mid-pregnancy (Fig. 2*A–C*). HFHS dams also ate fewer meals before pregnancy but more meals in late pregnancy compared to control dams (Fig. 2*D*). In control dams, food and caloric intake increased in early pregnancy followed by a further rise in mid-pregnancy, which was maintained in late pregnancy (Fig. 2*A* and *B*). This was primarily due to the consumption of larger meals with advancing pregnancy (Fig. 2*C*), without changes in meal number (Fig. 2*D*). In contrast, in HFHS dams, food and caloric intake increased only in late pregnancy, accompanied by larger meal size and increased meal number (Fig. 2*A–D*).

During the dark phase, food intake was lower but caloric intake similar in HFHS compared to control dams (Fig. 2*A* and *B*). Although food and caloric intake differed between pregnancy stages, these did not differ between any pairs of time points (Fig. 2*A* and *B*). Despite this, we observed widespread changes in meal size and number during pregnancy, which differed between groups (Interaction $P < 0.01$ for both, Fig. 2*C* and *D*). HFHS dams consumed smaller meals than controls in mid- and late pregnancy (Fig. 2*C*) and ate more meals than control dams in late pregnancy only (Fig. 2*D*). In control dams, meal size increased in late pregnancy while meal number was lower throughout pregnancy compared to pre-mating values (Fig. 2*C* and *D*). In contrast, in HFHS dams, meal size did not change during pregnancy and meal number was lower than pre-mating values only in early pregnancy (Fig. 2*C* and *D*).

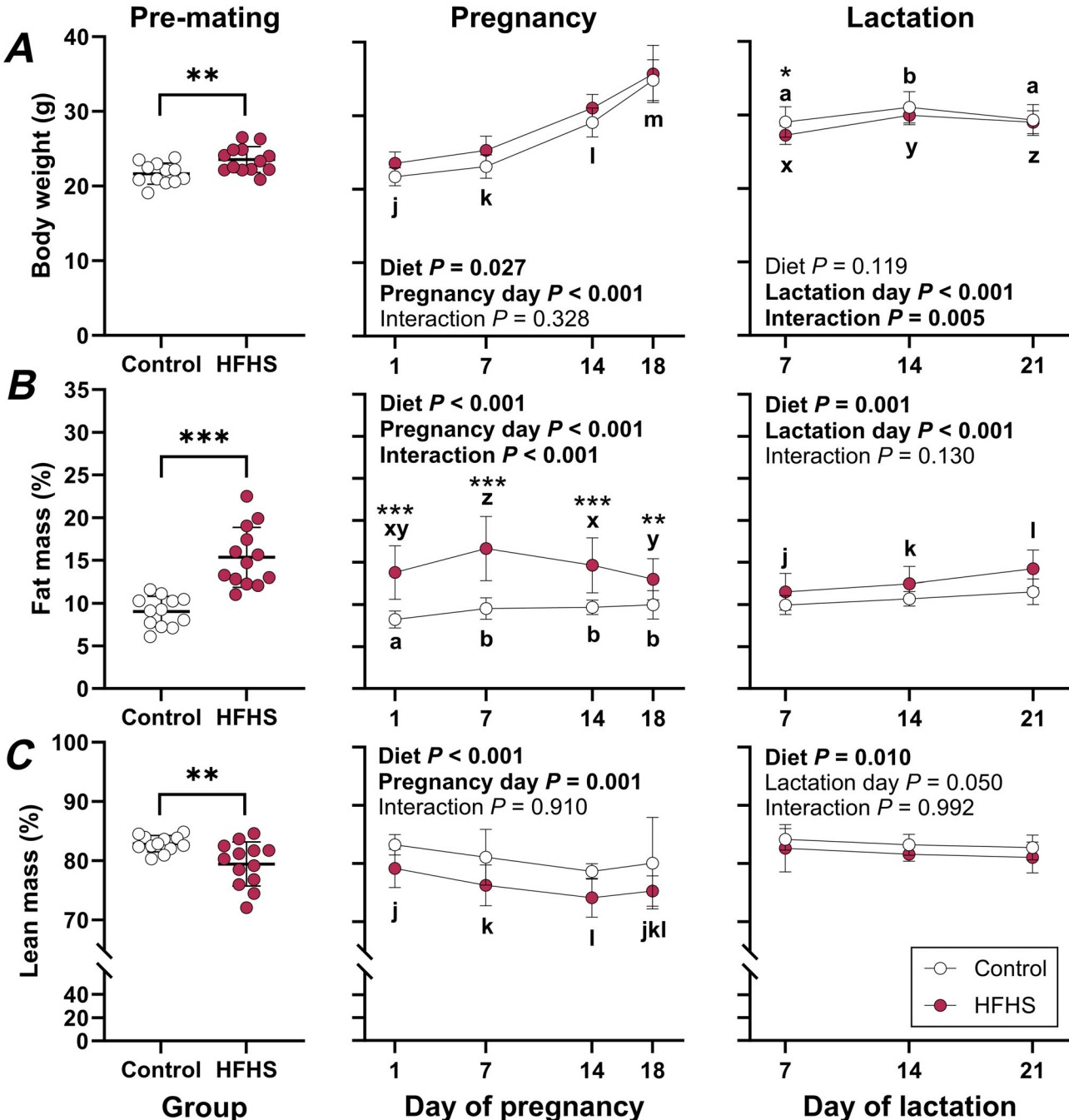

**Figure 1. Body weight (*A*), fat mass (*B*) and lean mass (*C*) in mice fed a control or high-fat, high-sugar (HFHS) diet for 11 weeks before mating and throughout pregnancy and lactation**

Because not all mice conceived and some dams were humanely killed for fetal studies, the groups studied pre-mating, throughout pregnancy and throughout lactation include different and partially overlapping subsets of female mice and were each analysed separately. Data for control mice are shown with open symbols and data for HFHS mice are shown with filled symbols. Pre-mating data (*n* = 12 Control, 13 HFHS mice) were compared by *t* test; symbols show individual values with lines and whiskers showing means and standard deviation for each group. Pregnancy (*n* = 13/group) and lactation data (*n* = 16 Control, 12 HFHS mice) were analysed separately by 2-way repeated measures ANOVA; symbols and whiskers show means and standard deviation at each day. Where interactions were significant, sub-analyses were performed to assess effects of diet at each day and effects of day within each diet group. Where day effects were significant, a Bonferroni comparison was used to compare pairs of days. Asterisks indicate differences between control and HFHS mice (*P < 0.05, **P < 0.01, ***P < 0.001). Differences between days are indicated with letters (overall: j, k, l, m; within control dams: a, b; within HFHS dams: x, y, z).

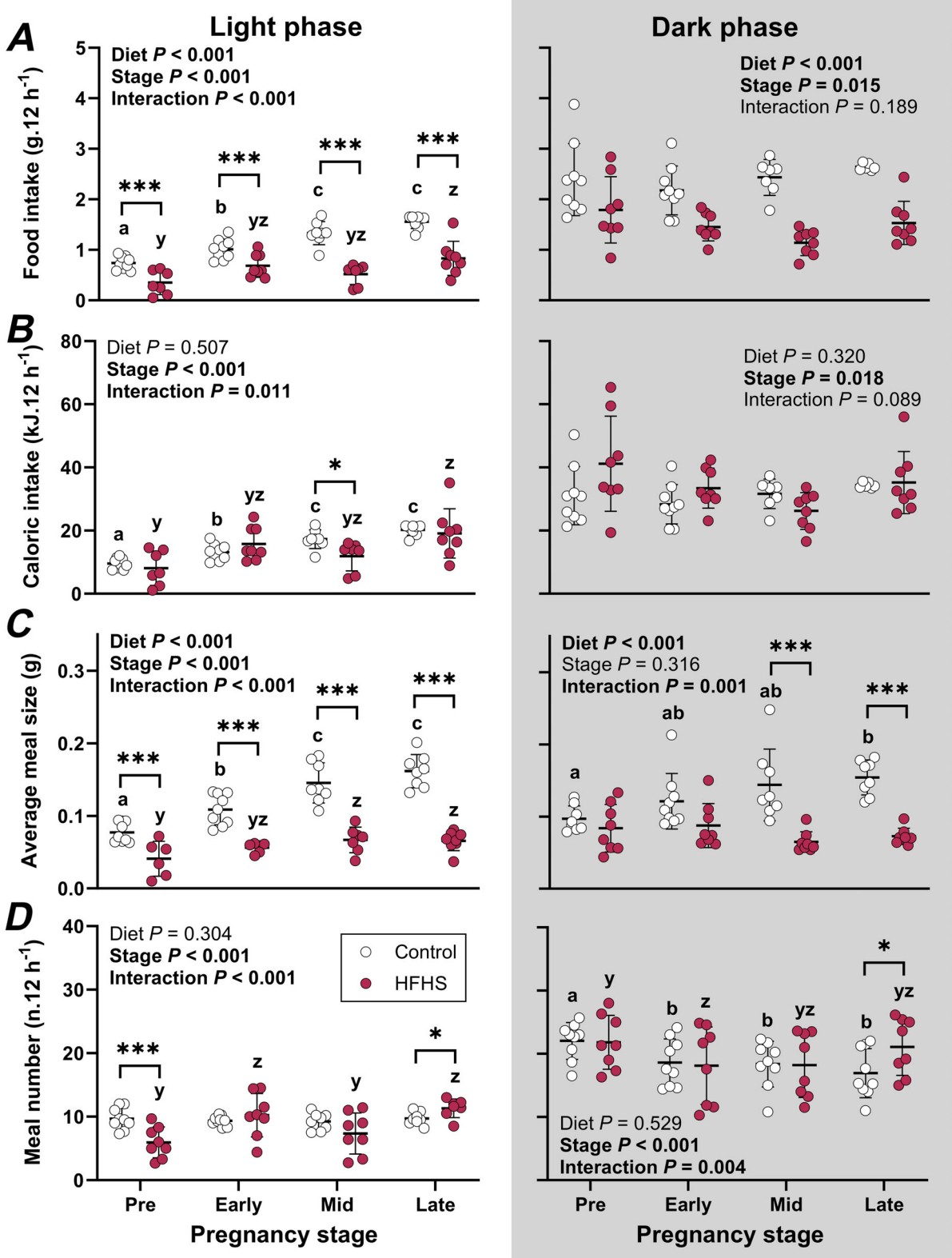

**Figure 2. Food intake (*A*), caloric intake (*B*), average meal size (*C*) and meal number (*D*) during a 12-h light and 12-h dark phase in mice fed a control or high-fat, high-sugar (HFHS) diet for 11 weeks before mating and throughout pregnancy and lactation**

Data were collected for 4 days prior to mating and from mating until gestational day 17.5 (*n* = 9 Control, 8 HFHS mice) and were analysed by 2-way repeated measures ANOVA; symbols show individual values with lines

### Greater energy expenditure in high-fat, high-sugar fed dams is accompanied by higher activity during the light but not the dark phase

During the light phase, both energy expenditure and activity were higher in HFHS dams compared to controls (Fig. 3A and B). Light phase energy expenditure steadily increased as pregnancy progressed, whilst activity remained stable before and during pregnancy. During the dark phase, energy expenditure was higher in HFHS than control dams, without differences in activity between diet groups (Fig. 3A and B). Dark phase energy expenditure differed between pregnancy stages, increasing from mid-pregnancy onwards (Fig. 3A). Activity during the dark phase also varied across pregnancy stages (Fig. 3B), falling by ~50% at early- and mid-pregnancy relative to pre-mating activity and decreasing further at late pregnancy.

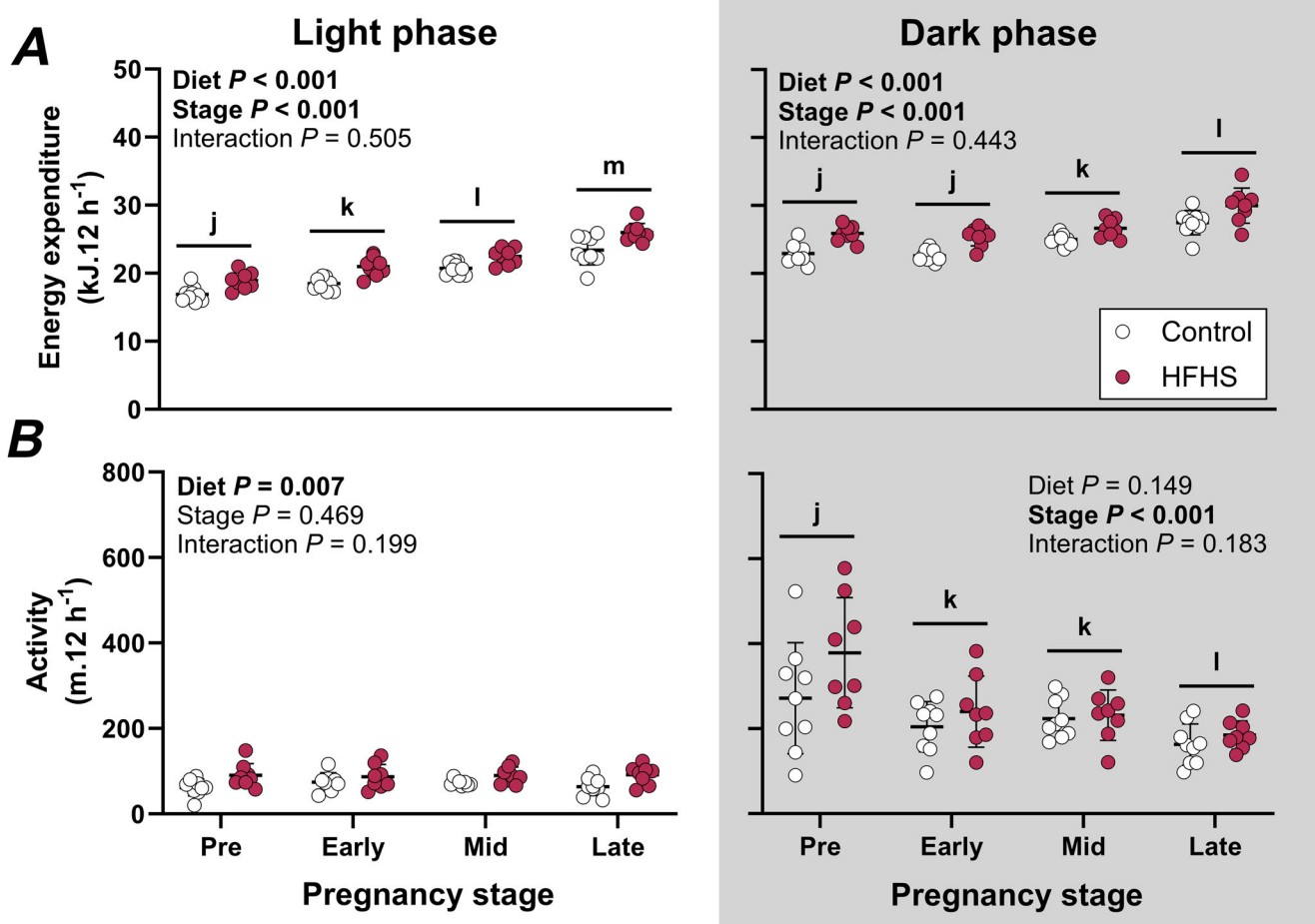

**Figure 3. Energy expenditure (A) and activity (B) during a 12-h light and 12-h dark phase in mice fed a control or high-fat, high-sugar (HFHS) diet for 11 weeks before mating and throughout pregnancy and lactation**

Data were collected for 4 days prior to mating and from mating until gestational day 17.5 (*n* = 9 Control, 8 HFHS mice) and were analysed by 2-way repeated measures ANOVA; symbols show individual values with lines and whiskers showing means and standard deviation for each group. Data for control mice are shown with open symbols and data for HFHS mice are shown with filled symbols. Where stage effects were significant, a Bonferroni comparison was used to compare pairs of stages. Differences between stages are indicated with letters (overall: j, k, l, m).

## Differential changes in fuel utilisation during pregnancy reflect diet composition

The changes in light phase respiratory exchange ratio (RER) and fat and carbohydrate oxidation during pregnancy differed between groups (Interaction $P < 0.001$ for all, Fig. 4A–C). RER was higher in control than HFHS dams during but not before pregnancy (Fig. 4A). Compared to controls, HFHS dams had higher fat oxidation before and throughout pregnancy whereas carbohydrate oxidation was lower in HFHS dams only in mid- and late pregnancy (Fig. 4B and C). In control dams, light phase RER increased in mid- and late pregnancy compared to pre-pregnancy (Fig. 4A), fat oxidation decreased in early pregnancy compared to pre-mating values (Fig. 4B) and carbohydrate oxidation increased steadily throughout pregnancy (Fig. 4C). In HFHS dams, light phase RER was higher in early- compared to mid-pregnancy (Fig. 4A) and fat oxidation was higher in mid- and late- compared to early pregnancy (Fig. 4B). Carbohydrate oxidation in HFHS dams increased in early pregnancy compared to pre-mating but remained unchanged throughout the rest of pregnancy (Fig. 4C). Effects of diet on dark phase RER and fat and carbohydrate oxidation also varied across stages (Interaction $P < 0.01$ for all, Fig. 4A–C). RER and carbohydrate oxidation were consistently lower, and fat oxidation was higher, in HFHS dams compared to controls (Fig. 4A–C). In control dams, RER and carbohydrate oxidation increased only in late pregnancy compared to pre-pregnancy values and fat oxidation remained stable (Fig. 4A–C). In HFHS dams, RER was increased only in early pregnancy (Fig. 4A), and fat and carbohydrate oxidation during pregnancy did not differ from pre-pregnancy (Fig. 4B and C).

## High-fat, high-sugar feeding impairs glucose tolerance prior to pregnancy, which worsens during pregnancy and lactation

Compared to control dams, HFHS dams had impaired glucose tolerance prior to mating, with elevated blood glucose from 30 to 90 min after a glucose challenge (Fig. 5A). This was accompanied by higher plasma insulin levels following a glucose challenge and higher blood glucose levels 10 min following an insulin challenge (Fig. 5B and C). At GD 16, HFHS dams remained glucose intolerant; however, plasma insulin concentration throughout the ipGTT was lower compared to controls (Fig. 5A and B). Moreover, the reduction in blood glucose concentration in response to an insulin challenge was comparable between the groups at GD 18 (Fig. 5C). At the end of lactation, glucose tolerance remained impaired in HFHS dams, with elevated fasting glucose as well as higher glucose concentrations throughout the ipGTT

(Fig. 5A). This was accompanied by higher plasma insulin concentrations before and following a glucose challenge (Fig. 5A and B). Blood glucose concentrations were also higher in HFHS dams compared to controls throughout the ipITT, but glucose area above the curve did not differ between diets (Fig. 5C). As expected, glucose tolerance, assessed as area under the glucose curve, was poorer during pregnancy compared to non-pregnant mice (Fig. 5A). Surprisingly, however, pregnancy was not accompanied by greater insulin secretion following a glucose challenge, assessed as insulin profile area above fasting insulin, nor a reduction in insulin sensitivity, assessed as glucose profile area above the curve following an insulin challenge (Fig. 5B and C).

## Maternal high-fat, high-sugar feeding has no impact on fetal growth but leads to fetal hyperglycaemia in late gestation

Despite HFHS dams having poor metabolic health, litter size, fetal weight and body size of offspring were not impacted by diet at GD 18 (Table 1). Accordingly, placental weight and efficiency were also comparable between maternal diet groups (Table 1). Overall, female fetuses were lighter, had shorter crown–rump and head length and smaller biparietal diameter compared to males. The weight and efficiency of female placentas were also less than males (Table 1). Consistent with the presence of maternal hyperglycaemia in HFHS dams, their fetuses had blood glucose levels that were on average 34% higher than control fetuses (Fig. 6A), although this was not accompanied by fetal hyperinsulinaemia (Fig. 6B). Blood glucose and plasma insulin levels were not different between fetal sexes (Fig. 6A,B).

## Offspring of high-fat, high-sugar fed dams are larger at PND 2

Despite similar body weight in neonates of control and HFHS dams at PND 2, the latter had greater crown–rump length, abdominal transverse diameter and head length (Table 2). In contrast to fetal life, body weight and size parameters did not differ between sexes at PND 2. Hyperglycaemia was no longer present in neonates from HFHS dams at PND 2 and plasma insulin concentrations were also comparable between groups (Table 2).

## Early postnatal growth is reduced in offspring of high-fat, high-sugar fed dams followed by catch-up growth and increased adiposity at weaning

At PND 7 and PND 14, body weights of male and female offspring of HFHS dams were lower compared

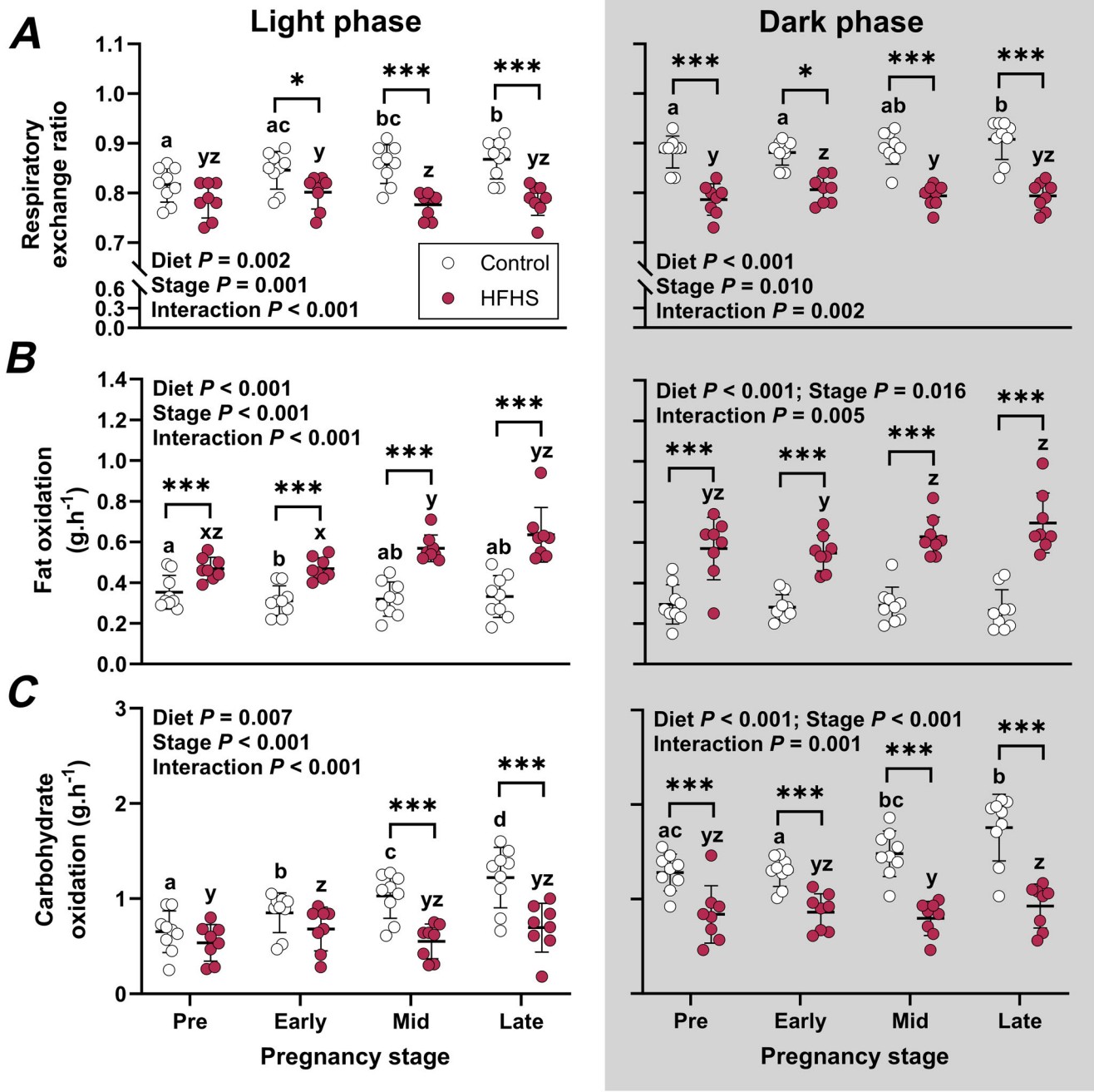

**Figure 4. Respiratory exchange ratio (*A*), fat oxidation (*B*) and carbohydrate oxidation (*C*) during a 12-h light and 12-h dark phase in mice fed a control or high-fat, high-sugar (HFHS) diet for 11 weeks before mating and throughout pregnancy and lactation**

Data were collected for 4 days prior to mating and from mating until gestational day 17.5 (*n* = 9 Control, 8 HFHS mice) and were analysed by 2-way repeated measures ANOVA; symbols show individual values with lines and whiskers showing means and standard deviation for each group. Data for control mice are shown with open symbols and data for HFHS mice are shown with filled symbols. Where interactions were significant, sub-analyses were performed to assess effects of diet at each stage and stage within each diet group. Where stage effects were significant, a Bonferroni comparison was used to compare pairs of stages. Asterisks indicate differences between control and HFHS mice (*$P$ < 0.05, ***$P$ < 0.001). Differences between stages are indicated with letters (within control dams: a, b, c, d; within HFHS dams: x, y, z).

to sex-matched controls (Fig. 7*A*). This was mediated by slower growth in the first but not the second week of life (Fig. 7*B*). In both sexes, offspring of HFHS dams then experienced catch-up growth in the week before weaning (Fig. 7*B*). Thus, at weaning, body weight was no longer different between groups (Fig. 7*A*). Offspring of HFHS dams had greater relative fat mass but similar relative lean mass at weaning compared to controls (Fig. 7*C*).

## Discussion

By conducting a detailed study of maternal physiology from before and throughout pregnancy and lactation, we have demonstrated that mice fed a HFHS diet provide a pre-clinical model of impaired glucose tolerance and elevated adiposity that persists from before mating and throughout pregnancy and lactation. This reflects

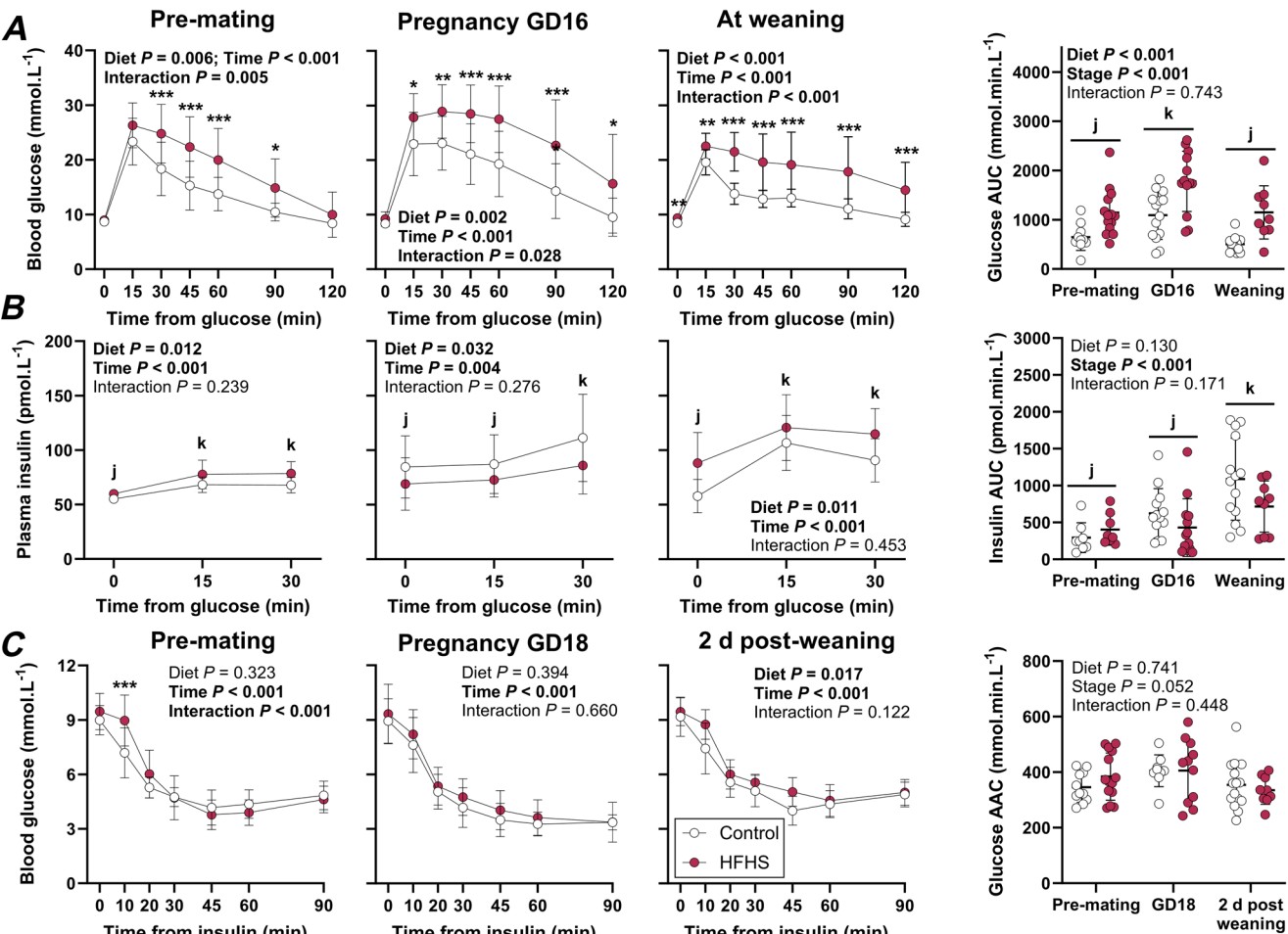

**Figure 5. Blood glucose (*A*) and plasma insulin (*B*) profiles and areas under the curve throughout an intraperitoneal glucose tolerance test, and blood glucose (*C*) profiles and area above the curve throughout an intraperitoneal insulin tolerance test in mice fed a control or high-fat, high-sugar (HFHS) diet for 11 weeks before mating and throughout pregnancy and lactation**
Blood glucose (*A*) and plasma insulin (*B*) concentrations before and after intraperitoneal glucose administration were analysed by 2-way repeated measures ANOVA to assess the impact of diet and time separately for data collected before mating, in late pregnancy and after weaning of offspring; symbols and whiskers show means and standard deviation ($n$ = 10–15 Control, 10–14 HFHS mice at each stage). Data for control mice are shown with open symbols and data for HFHS mice are shown with filled symbols. Blood glucose concentrations (*C*) before and after intraperitoneal insulin administration were analysed by 2-way repeated measures ANOVA to assess the impact of diet and time separately for data collected before mating, in late pregnancy and after weaning of offspring; symbols and whiskers show means and standard deviation ($n$ = 9–15 Control, 9–14 HFHS mice at each stage). Asterisks indicate differences between control and HFHS mice (*$P$ < 0.05, **$P$ < 0.01, ***$P$ < 0.001). Where time effects were significant, a Bonferroni comparison was used to compare pairs of times. Differences between times are indicated with letters (overall; j, k). Areas under *A*, *B* and above *C* the curve were analysed by 2-way ANOVA to assess effects of diet and stage; symbols show individual values with lines and whiskers showing means and standard deviation for each group. Where stage effects were significant, a Bonferroni comparison was used to compare pairs of stages. Differences between stages are indicated with letters (overall: j, k).

**Table 1. Fetal weight and size and placental weight and efficiency of control and high-fat, high-sugar (HFHS) fed mice at GD 18**

| | Control | | HFHS | | Significance | | | |
| --- | --- | --- | --- | --- | --- | --- | --- | --- |
| | Male | Female | Male | Female | Maternal diet | Fetal sex | Interaction | Litter size |
| Fetal size | | | | | | | | |
| Body weight (g) | 0.862 ± 0.140 (37) | 0.876 ± 0.146 (33) | 0.862 ± 0.166 (37) | 0.790 ± 0.157 (30) | 0.442 | **0.002** | 0.748 | 0.069 |
| Crown–rump length (mm) | 20.5 ± 1.6 (36) | 20.3 ± 1.6 (31) | 21.1 ± 1.7 (31) | 19.9 ± 1.6 (26) | 0.640 | **0.014** | 0.764 | 0.820 |
| Abdominal transverse diameter (mm) | 8.2 ± 0.7 (37) | 8.4 ± 0.6 (31) | 8.2 ± 0.7 (31) | 7.7 ± 0.6 (26) | 0.290 | 0.905 | 0.453 | 0.263 |
| Head length (mm) | 9.6 ± 0.8 (37) | 9.7 ± 0.6 (31) | 9.7 ± 0.7 (31) | 9.2 ± 0.6 (26) | 0.441 | **0.010** | 0.514 | 0.640 |
| Biparietal diameter (mm) | 3.8 ± 0.2 (36) | 3.7 ± 0.3 (31) | 4.0 ± 0.2 (31) | 3.9 ± 0.3 (25) | **0.004** | **0.004** | 0.604 | 0.316 |
| Placental measures | | | | | | | | |
| Placental weight (g) | 0.087 ± 0.009 (34) | 0.079 ± 0.012 (32) | 0.090 ± 0.016 (33) | 0.077 ± 0.021 (25) | 0.457 | **< 0.001** | 0.816 | 0.059 |
| Placental efficiency | 10.0 ± 2.0 (34) | 11.3 ± 2.5 (32) | 10.1 ± 2.8 (33) | 11.5 ± 3.4 (24) | 0.758 | **< 0.001** | 0.289 | 0.826 |

Effects of maternal diet and fetal sex were analysed by mixed model, treating measures of individual pups as repeated measures on the dam, and including litter size as a covariate in analyses. Results are presented as means ± SD (*n* fetuses).

**Table 2. Offspring weight and size and metabolic measures of control and high-fat, high-sugar (HFHS) fed mice at PND 2**

| | Control | | HFHS | | Significance | | |
| --- | --- | --- | --- | --- | --- | --- | --- |
| | Male | Female | Male | Female | Maternal diet | Pup sex | Interaction |
| Pup size | | | | | | | |
| Body weight (g) | 1.50 ± 0.10 (17) | 1.49 ± 0.24 (17) | 1.50 ± 0.23 (11) | 1.38 ± 0.19 (11) | 0.155 | 0.515 | 0.528 |
| Crown–rump length (mm) | 25.33 ± 1.72 (12) | 25.89 ± 1.33 (11) | 26.98 ± 1.71 (13) | 26.43 ± 1.86 (10) | **0.033** | 0.991 | 0.268 |
| Abdominal transverse diameter (mm) | 8.70 ± 0.61 (12) | 8.46 ± 0.60 (11) | 8.99 ± 0.50 (13) | 8.91 ± 0.52 (10) | **0.031** | 0.954 | 0.217 |
| Head length (mm) | 11.20 ± 0.69 (12) | 10.89 ± 1.16 (11) | 11.33 ± 0.84 (13) | 11.87 ± 0.60 (10) | **0.032** | 0.651 | 0.099 |
| Biparietal diameter (mm) | 4.59 ± 0.30 (12) | 4.34 ± 0.46 (11) | 4.59 ± 0.52 (13) | 4.57 ± 0.57 (10) | 0.410 | 0.340 | 0.407 |
| Metabolic measures | | | | | | | |
| Blood glucose (mmol L$^{-1}$) | 4.72 ± 0.90 (22) | 4.66 ± 0.98 (22) | 4.34 ± 1.10 (18) | 4.66 ± 0.91 (13) | 0.418 | 0.585 | 0.412 |
| Plasma insulin (pmol L$^{-1}$) | 36.8 ± 24.8 (21) | 31.5 ± 38.4 (19) | 33.0 ± 26.1 (12) | 46.2 ± 41.8 (8) | 0.938 | 0.648 | 0.712 |

Effects of maternal diet and pup sex were analysed by 2-way ANOVA, with measures made on no more than one female and one male per litter. Results are presented as means ± SD (*n*).

the clinical situation of women entering pregnancy overweight, obese or with impaired glucose tolerance rather than pregnancy-induced metabolic dysfunction. Our findings of pre-pregnancy maternal metabolic dysfunction reinforce the importance of longitudinal studies in each model to define the extent and timing of offspring exposures, and likely consequences for offspring development.

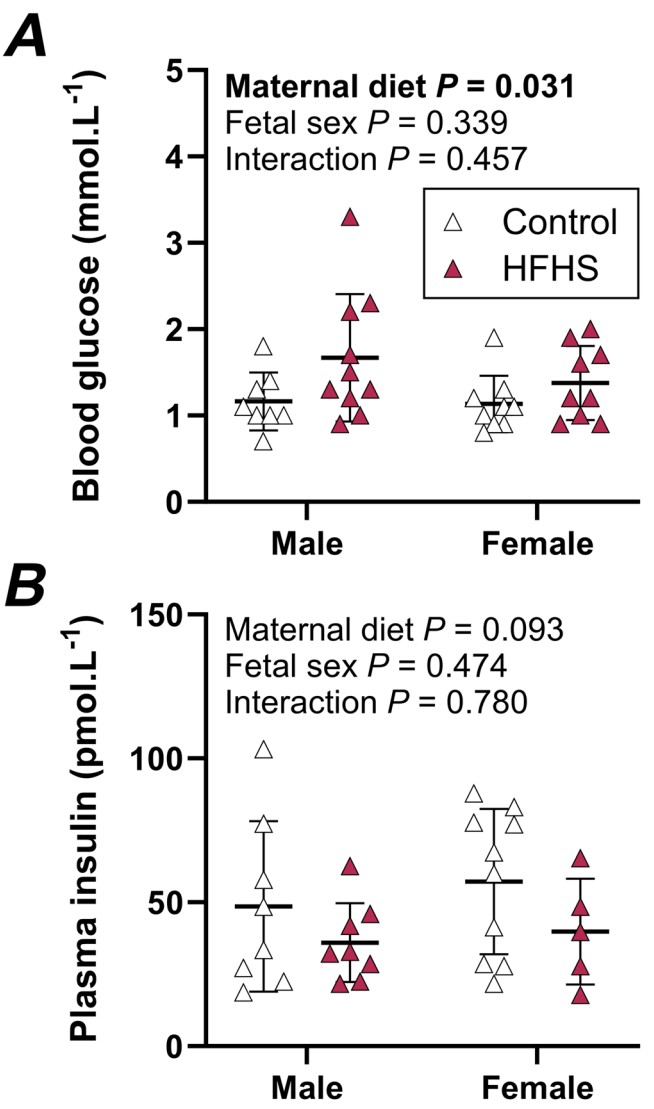

**A**

**Maternal diet *P* = 0.031**
Fetal sex *P* = 0.339
Interaction *P* = 0.457

Blood glucose (mmol.L$^{-1}$)

△ Control
▲ HFHS

Male   Female

**B**

Maternal diet *P* = 0.093
Fetal sex *P* = 0.474
Interaction *P* = 0.780

Plasma insulin (pmol.L$^{-1}$)

Male   Female

**Figure 6. Fetal blood glucose (*A*) and plasma insulin (*B*) concentrations of control and high-fat, high-sugar (HFHS) fed mice at GD 18**
Data were analysed by 2-way ANOVA to assess effects of maternal diet and fetal sex; symbols show individual values with lines and whiskers showing means and standard deviation for each group. Data for control mice are shown with open symbols and data for HFHS mice are shown with filled symbols. Within each sex, each fetus was sampled from a different litter; *n* = 8 control males, 9–10 control females, 8–10 HFHS males, 5–9 HFHS females.

Consistent with previous studies, the body mass of mice fed a control diet increased throughout pregnancy, which was supported by increased food and caloric intake during the light phase, and decreased activity during the dark phase (Clarke et al., 2024; Fekete, 1954; Lean et al., 2022; Mort et al., 2023; Vega et al., 2015). In this study we assessed longitudinal changes in substrate oxidation for the first time in mouse pregnancy. Fuel usage in our control mice changed throughout pregnancy, with increases in light phase RER evident from mid-pregnancy and in dark phase RER at late pregnancy. Increasing RER throughout pregnancy reflected increasing carbohydrate oxidation to meet energy requirements, as occurs in humans (Butte et al., 2004; Melzer et al., 2014). The early pregnancy reduction in fat oxidation in the present study, despite increasing energy expenditure, is consistent with suppression of lipolysis early in human pregnancy to allow fat storage for oxidation in late gestation and lactation (Rebuffé-Scrive et al., 1985). It is not clear whether the subsequent normalisation of fat oxidation at mid-pregnancy in our mice reflects an increase in lipolysis, which remains suppressed through the first two trimesters of human pregnancy (Alvarez et al., 1996; Villar et al., 1992).

We also investigated the impacts of normal pregnancy and lactation on maternal metabolism. In the present study, glucose tolerance was impaired in control mice in late pregnancy (GD 16) compared to pre-mating and normalised by weaning, consistent with the impaired glucose tolerance reported previously in pregnant mice at GD 11 (Drynda et al., 2015). In contrast, others have reported similar glucose tolerance in late pregnant and non-pregnant control mice (Makarova et al., 2010; Musial et al., 2016). Blood glucose profiles after oral glucose administration remain similar throughout pregnancy in most litter-bearing species (Overduin et al., 2025). In humans, however, the rise in blood glucose becomes delayed and peak glucose concentrations increase with advancing pregnancy (Overduin et al., 2025). These species differences may reflect differences in total fetal demand and placental uptake. Upregulation of insulin secretion and $\beta$-cell mass appears to be common to murine (Beamish et al., 2017; Drynda et al., 2015; Szlapinski et al., 2019; Zhang et al., 2022) and human pregnancy (Baeyens et al., 2016), although acute insulin responses to I.P. glucose did not differ between non-pregnant and late pregnant mice in the present study. Normalisation of glucose tolerance during lactation, which in some studies is also improved relative to non-pregnant controls (Canul-Medina et al., 2021; Moon et al., 2020; Vicente et al., 2020), may reflect mammary uptake of glucose for milk secretion and increases in maternal $\beta$-cell mass and glucose-stimulated insulin secretion. It is unclear why whole-body insulin sensitivity assessed using an ipITT was similar before

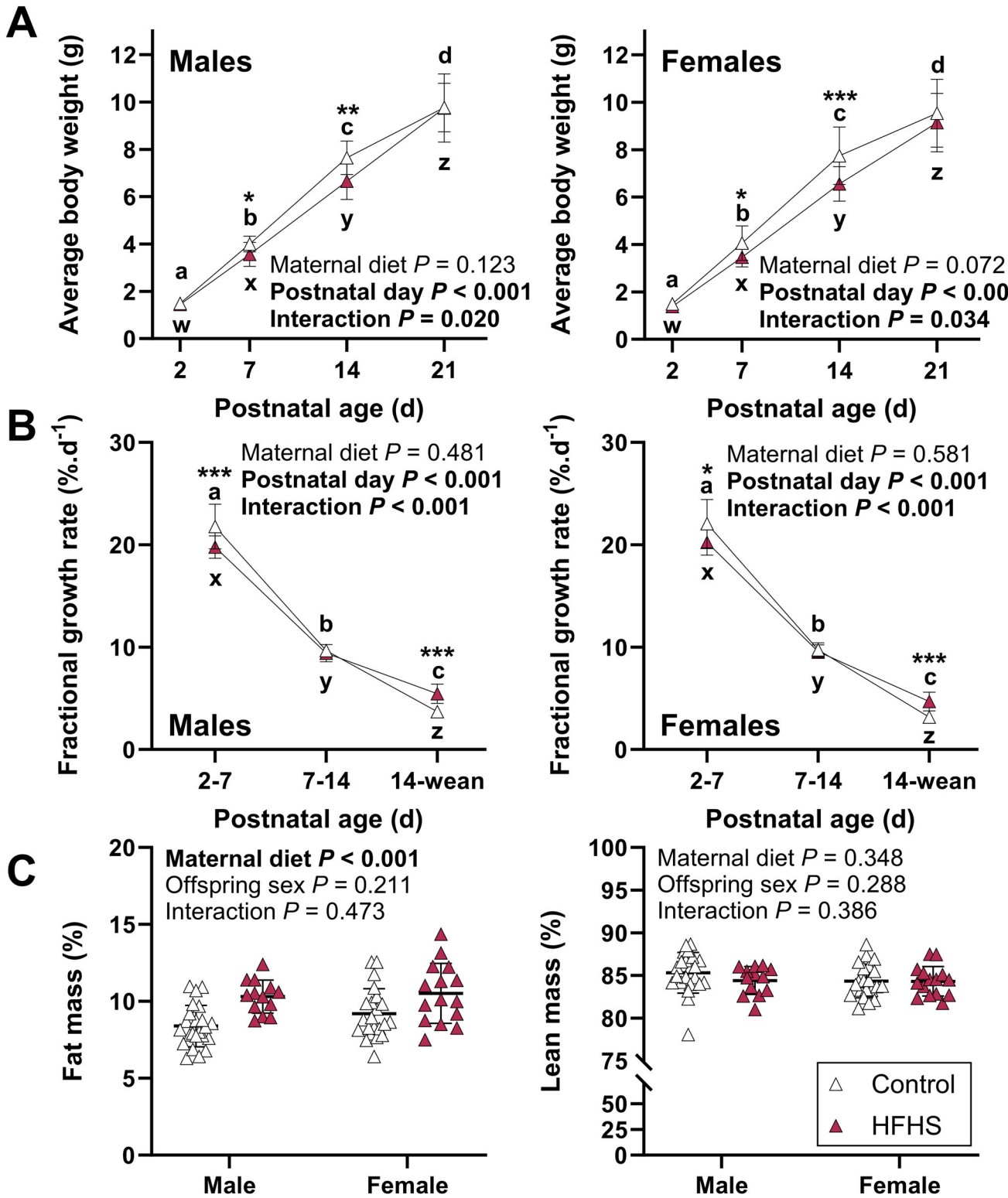

**Figure 7. Litter-average body weight (*A*) and fractional growth rate (*B*) from postnatal day 2 until weaning, and body composition at weaning (*C*) in male and female offspring of mice fed a control or high-fat, high-sugar (HFHS) diet for 11 weeks before mating and throughout pregnancy and lactation**
Effects of maternal diet and postnatal age on body weight and fractional growth rate were analysed separately for male (control: *n* = 17, HFHS: *n* = 11 litters) and female (control: *n* = 16, HFHS: *n* = 11 litters) offspring by 2-way repeated measures ANOVA; symbols and whiskers show means and standard deviation at each age. Data

for control mice are shown with open symbols and data for HFHS mice are shown with filled symbols. Where interactions were significant, sub-analyses were performed to assess effects of diet at each age and effects of age within each diet group. Where age effects were significant, a Bonferroni comparison was used to compare pairs of ages. Asterisks indicate differences between control and HFHS mice (*$P < 0.05$, **$P < 0.01$, ***$P < 0.001$). Differences between ages are indicated with letters (within offspring of control dams: a, b, c, d; within offspring of HFHS dams: w, x, y, z). Fat and lean mass data were analysed by 2-way ANOVA to assess effects of maternal diet and offspring sex; symbols show individual values with lines and whiskers showing means and standard deviation for each group.

pregnancy, at GD 18 and shortly after weaning in the present study. Most studies using ipITT to assess insulin sensitivity during normal mouse pregnancy have reported reduced insulin sensitivity in mid–late pregnant mice (Drynda et al., 2015; Makarova et al., 2010; Musial et al., 2016), although not at GD 7 (Makarova et al., 2010), compared to non-pregnant controls. Whole-body insulin sensitivity measured as glucose infusion rate (GIR) during a hyperinsulinaemic euglyaecemic clamp was also lower in mid-pregnant than non-pregnant mice, but was similar to non-pregnant values at GD 19 (Mishra et al., 2022; Musial et al., 2016).

Although the present study allowed us to define these adaptations to pregnancy in control mice, our primary purpose was to investigate the impacts of feeding mice a HFHS diet on maternal physiology. Feeding female mice, a HFHS diet for 10–11 weeks led to modestly higher body weight (↑9%) and higher adiposity (↑70%) prior to mating, compared to control mice. The degree to which HFHS diets increase maternal body weight (3-73%) and/or adiposity (128–208%) at the start of pregnancy are highly variable (Fernandez-Twinn et al., 2017; Hufnagel et al., 2022; Lean et al., 2022; Loche et al., 2018; Mort et al., 2023; Napso et al., 2022; Nicholas & Ozanne, 2019). Studies employing longer pre-mating feeding durations (≥10 weeks) and highly palatable diets, such as those supplemented with condensed milk to promote hyperphagia, typically induce overt obesity, with dams often 10 g heavier and with 3-fold more body fat at mating than controls (Fernandez-Twinn et al., 2017; Hufnagel et al., 2022; Loche et al., 2018). Differences in body weight and adiposity of HFHS and control mice in the present study are similar to protocols using shorter HFHS feeding durations of 1–3 weeks prior to mating (Lean et al., 2022; Mishra et al., 2022). The HFHS diet used in the present study did not induce persistent hyperphagia, with similar energy intakes in HFHS and control mice in the week before mating. Effects of HFHS on appetite were not observed until pregnancy, when HFHS-fed dams consumed less food overall, with delayed pregnancy-induced increases in food and caloric intake compared to controls. Reduced gestational food intake in non-obese HFHD-fed pregnant mice is consistent with previous reports (Clarke et al., 2025; Mort et al., 2023). Despite their lower food intake, we observed similar pregnancy weight gains in HFHS-fed and control dams, although HFHS-feeding was associated with lower pregnancy weight gain in some other studies (Lean et al., 2022; Mort et al., 2023; Napso et al., 2022). This may reflect different impacts on energy intake, with reductions throughout pregnancy in some studies (Mort et al., 2023), whereas we observed lower energy intake in the HFHS group only at mid-pregnancy. The decrease in body fat percentage that we observed in HFHS-fed but not control-fed mice from mid-pregnancy is consistent with a previous report (Lean et al., 2022). Altered fat deposition likely reflects altered fuel usage and may be a response to the diet rather than its impacts on maternal metabolism or adiposity (Bhandarkar et al., 2021), since reduced fat gain in late pregnancy was observed even when HFHS feeding only started from conception (Musial et al., 2017). Consistent with altered fuel usage in HFHS-fed mice, their fat oxidation increased from mid-pregnancy, in contrast to control mice, which exhibited the expected shift from fat to carbohydrate utilisation with advancing pregnancy (Butte et al., 2004; Melzer et al., 2014). An additional consequence of the reduced food intake induced by HFHS feeding in rodents is reduced protein intake throughout pregnancy (Mort et al., 2023), which itself impacts maternal and offspring health (Jahan et al., 2020). In contrast to pregnancy, when HFHS-fed mice were heavier than controls, the HFHS mice were lighter than controls by the end of the first week of lactation. Although we did not house lactating mice in metabolic cages and therefore do not have measures of lactation food intake and metabolism in the present study, others have reported that HFHS-feeding reduced caloric as well as food and protein intakes throughout lactation (Mort et al., 2023). HFHS-feeding may thus impair the normal upregulation of appetite during lactation, resulting in greater catabolism of reserves.

A strength of the present study is our characterisation of maternal insulin-regulated glucose metabolism at multiple time points. Prior to mating, female mice fed a HFHS diet for 10–11 weeks had similar fasting glucose, glucose-stimulated insulin secretion and insulin sensitivity as control mice. However, even at this stage, these mice had impaired glucose tolerance, elevated circulating insulin concentrations and a delayed fall in circulating glucose after insulin administration. It seems likely that impaired glucose tolerance and/or whole-body insulin resistance are consequences of increased maternal

adiposity. These metabolic impairments are consistently reported in HFHS dams with greater pre-mating weight gains than controls (Carter et al., 2015; Huypens et al., 2016; King et al., 2013; Mishra et al., 2022; Sasson et al., 2015; Sissala et al., 2022), but insulin sensitivity is not impaired in mice fed HFHS for a single week (Mishra et al., 2022). In contrast, although we did not see evidence of an enhanced insulin response to a glucose challenge, the higher insulin concentrations in HFHS mice before and throughout glucose challenge suggest increased insulin secretion overall. Maternal insulin secretion may be upregulated by a HFHS diet independent of weight gain, since *ex vivo* islet basal and glucose-stimulated insulin secretion was higher in HFHS than control diet-fed mice after only a week (Mishra et al., 2022). HFHS diet-induced stimulation of insulin secretion capacity persists with longer diet exposure in non-pregnant female mice (Huypens et al., 2016; King et al., 2013; Sissala et al., 2022).

Impaired insulin-regulated glucose metabolism in HFHS dams persisted during pregnancy in the present study, including impaired glucose tolerance at GD 16, although not fasting hyperglycaemia. This is consistent with the observation that HFHS diet feeding begun at least 3 weeks before mating induced impaired glucose tolerance during pregnancy in most (Carter et al., 2015; Fernandez-Twinn et al., 2017; Hufnagel et al., 2022; Park et al., 2020; Rosario et al., 2015; Sissala et al., 2022), but not all other studies (King et al., 2013; Lean et al., 2022), with the latter potentially reflecting small sample sizes. Impacts of HFHS feeding commencing at, or in the week prior to, mating were more variable (Musial et al., 2017; Pennington et al., 2017) and in some cases impaired glucose tolerance emerged later in pregnancy than in the present study (Musial et al., 2017). In the present study, HFHS dams had lower circulating insulin concentrations, but similar insulin responses to a glucose challenge, than control dams at GD 16. Others have reported lower *in vivo* glucose-stimulated insulin secretion in HFHS-fed pregnant mice (Pennington et al., 2017), likely reflecting impairment of both $\beta$-cell function (Mishra et al., 2022) and pregnancy-induced $\beta$-cell expansion (Lean et al., 2022; Mishra et al., 2022; Pennington et al., 2017). However, this impaired pregnancy-induced upregulation in HFHS-fed mice does not always result in lower insulin secretion capacity. In mice where HFHS feeding has already induced insulin resistance and compensatory upregulation of insulin secretion before pregnancy, insulin secretion during pregnancy may remain elevated relative to controls (Hufnagel et al., 2022; King et al., 2013; Rosario et al., 2015). Similar to our pre-mating findings, insulin sensitivity at late pregnancy did not differ between diet groups in the present study. This is consistent with a lack of effect of HFHS-feeding on whole-body insulin sensitivity in late pregnancy reported by others when diets commenced at mating (Musial et al., 2017), but contrasts with insulin resistance at GD 13.5 in mice that started HFHS feeding a week before mating (Mishra et al., 2022), and in late pregnant mice after 17 weeks of HFHS feeding (Hufnagel et al., 2022). However, the lack of change in whole-body insulin sensitivity in the present study does not necessarily indicate that HFHS feeding has not altered maternal metabolism. Even with unchanged whole-body insulin sensitivity, insulin sensitivity and expression of insulin signalling pathway proteins were higher in liver, but lower in skeletal muscle and white adipose tissue, of HFHS compared to control dams at late pregnancy (Musial et al., 2017).

Our measures in fetuses and neonatal offspring suggest that maternal HFHS-feeding disrupts fetal metabolism and early postnatal growth, with similar effects in male and female fetuses and neonates. Few studies have reported sex-specific outcomes, with inconsistent and limited evidence for sex-specific impacts of a maternal HFHSD. Lean & coauthors (2022) did not observe differences in fetal or placental weight between HFHS and control pregnancies in either sex, concurrent with elevated circulating glucose and insulin at GD 18.5 in both sexes. Interestingly, changes in placental structure that reduced placental specific diffusion capacity were observed in male placentas only (Lean et al., 2022). In models of frank maternal obesity, while Hufnagel & co-authors (2022) reported that male and female pregnancies had similarly higher placental weights and lower fetal weights, placental efficiency and placental labyrinth proportion in late gestation, King & coauthors (2013) observed reduced birthweights in females only. In the present study, litter size, fetal weight, most measures of fetal size and placental weight and efficiency were comparable in fetuses of HFHS and control dams at GD17.5. However, fetuses of HFHS dams had higher blood glucose concentrations than controls, consistent with previous reports (Lean et al., 2022; Napso et al., 2022). Fetal hyperglycaemia may be a consequence of impaired maternal glucose tolerance leading to increased glucose supply via facilitated diffusion of glucose across the placenta, which occurs down a concentration gradient (Lasunción et al., 1987). Differences in placental structure and nutrient transporter expression likely also contribute to higher glucose concentrations in the HFHS compared to control fetal mouse. Fetal and placental glucose uptake are higher in HFHS dams compared to controls at GD 13.5, concurrent with elevated maternal circulating glucose concentrations and reduced glucose uptake by maternal tissues (Mishra et al., 2022). At GD 15.5 transfer across the placenta and fetal accumulation of glucose and neutral amino acid were also greater in HFHS than control dams (Sferruzzi-Perri et al., 2013). These differences were not observed at GD 18.5 (Sferruzzi-Perri et al., 2013) despite greater surface area for exchange at

both ages, possibly reflecting greater expression of glucose and amino acid transporters at GD 15.5 but not GD 18.5 (Sferruzzi-Perri et al., 2013). Despite elevated circulating glucose in HFHS fetuses at GD 17.5, circulating insulin was similar in both groups in the present study. This likely reflects the immature status of fetal mouse $\beta$-cells, in which insulin secretion does not increase in response to elevated glucose (Blum et al., 2012). At GD 18.5, Lean and coauthors reported higher circulating insulin concentrations in HFHS than control fetuses, which might reflect premature gain of glucose-responsiveness in $\beta$-cells or subtle differences between models (Lean et al., 2022).

At 2 days of age, we did not see differences in circulating glucose or insulin in offspring, potentially reflecting cessation of the elevated placental glucose supply. The impacts of maternal HFHS exposure *in utero* and throughout lactation on postnatal $\beta$-cells have yet to be investigated. Nevertheless, impaired offspring metabolic health in adulthood, including impaired glucose tolerance, insulin resistance and altered glucose-stimulated insulin secretion, is a common outcome following maternal HFHS-feeding, even when offspring fetal or birth weights are unchanged (Nicholas et al., 2016; O'Hara et al., 2021). Offspring growth patterns showed greater changes postnatally, with several markers of fetal size greater in 2-day-old neonates of HFHS than control dams. HFHS offspring grew slower than control offspring in the first week of life, followed by catch-up growth in the third week, mirroring the early postnatal growth trajectory in a similar HFHS cohort (Mort et al., 2023). It is not clear whether altered growth patterns before weaning reflect *in utero* programming or altered milk production or composition (Lean et al., 2022). Despite similar weights, offspring of HFHS dams were fatter than control offspring at weaning, consistent with a previous report (Casasnovas et al., 2021).

In conclusion, consumption of a HFHS diet before and throughout pregnancy and lactation in mice results in impaired glucose metabolism, which progressively deteriorates across pregnancy and lactation, including development of fasting hyperglycaemia and failure of circulating glucose concentrations to return to baseline. Consumption of the HFHS diet also altered the pattern of metabolic adaptations to pregnancy, including a failure to shift from fat to carbohydrate oxidation, reduced fat deposition across pregnancy and failure of pancreatic $\beta$-cells to compensate for pregnancy-associated insulin resistance. These maternal effects were associated with fetal hyperglycaemia, without hyperinsulinaemia and altered neonatal growth patterns of offspring, which were fatter at weaning. Consequences of HFHS-induced changes to the offspring environment *in utero* and throughout lactation for the metabolic health of offspring, including their $\beta$-cell function, are the subject of ongoing studies.

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

## Additional information

### Data availability statement

The datasets generated during the current study are available from Dr. Lisa M. Nicholas upon reasonable request.

### Competing interests

The authors have no conflicts of interest.

## Author contributions

L.M.N. conceived and designed research. S.E.O., K.M.G., G.S.C., A.J.P., K.L.G. and L.M.N. performed experiments. L.M.N., K.L.G., G.S.C. and A.M.J. analysed data. L.M.N., K.L.G., S.E.O. and A.M.J. interpreted results of experiments. S.E.O., K.L.G. and L.M.N. drafted manuscript. All authors edited and revised the manuscript. All authors have read and approved the final version of this manuscript and agree to be accountable for all aspects of the work in ensuring that questions related to the accuracy or integrity of any part of the work are appropriately investigated and resolved. All persons designated as authors qualify for authorship, and all those who qualify for authorship are listed.

## Funding

This work was supported by a fellowship from the National Health and Medical Research Council (GNT1092158) and a lab establishment grant from the Faculty of Health and Medical Research and the Robinson Research Institute at the University of Adelaide and SAHMRI to L.M.N. This work was also supported by a University of Adelaide, Faculty of Health and Medical Sciences Strategic Grant to A.J.P. S.E.O is funded by a University of Adelaide Research Training Scholarship. G.S.C. was funded by an Australian Government Research Training Program Scholarship.

## Acknowledgements

We thank Professors Frank Reimann and Fiona Gribble from the Wellcome-MRC Institute of Metabolic Science-Metabolic Research Laboratories (Cambridge, UK) for provision of the genetic mouse line. We also thank staff at SAHMRI Bioresources for their invaluable assistance with the care and management of the animals used in this study.

Open access publishing facilitated by Adelaide University, as part of the Wiley - Adelaide University agreement via the Council of Australasian University Librarians.

## Keywords

glucose, high-fat, high-sugar diet, insulin, lactation, metabolism, mouse, pregnancy

## Supporting information

Additional supporting information can be found online in the Supporting Information section at the end of the HTML view of the article. Supporting information files available:

**Peer Review History**

