## [Peer Review History · The Journal of Physiology]

A high-fat, high-sugar diet impairs maternal metabolism throughout pregnancy and lactation in mice

Stephanie Elise O'Hara, Kelly M Gembus, Georgia S Clarke, Amanda J Page, Kathryn L Gatford, and Lisa Marie Nicholas
DOI: 10.1113/JP289985

Corresponding author(s): Lisa Nicholas (lisa.nicholas@adelaide.edu.au)

Review Timeline:

Submission Date:	31-Aug-2025
Editorial Decision:	13-Oct-2025
Revision Received:	07-Dec-2025
Editorial Decision:	02-Jan-2026
Revision Received:	13-Jan-2026
Accepted:	20-Jan-2026

Senior Editor: Laura Bennet

Reviewing Editor: Bruno Grassi

Transaction Report:

Re: JP-RP-2025-289985 "**A high-fat, high-sugar diet impairs maternal metabolism throughout pregnancy and lactation in mice**" by Stephanie Elise O'Hara, Kelly M Gembus, Georgia S Clarke, Amanda J Page, Kathryn L Gatford, and Lisa Marie Nicholas

Dear Dr Nicholas,

Thank you for submitting your manuscript to The Journal of Physiology. It has been assessed by a Reviewing Editor and by 2 expert referees and we are pleased to tell you that it is potentially acceptable for publication following satisfactory major revision.

Please address all the points raised and incorporate all requested revisions or explain in your Response to Referees why a change has not been made. We hope you will find the comments helpful and that you will be able to return your revised manuscript within 2 months. If your article is NOT for a Special Issue, you may have 9 months to revise. If you require an extension, please contact journal staff: jp@physoc.org. Please note that this letter does not constitute a guarantee for acceptance of your revised manuscript.

REVISION CHECKLIST:

We look forward to receiving your revised submission.

Yours sincerely,

Laura Bennet
Senior Editor
The Journal of Physiology

REQUIRED ITEMS

- Please upload separate high-quality figure files via the submission form.

EDITOR COMMENTS

Reviewing Editor:

Comments to the Author:

The aim of the study is to evaluate, in a mouse model, the effect of a high-fat, high-sugar diet during pregnancy and lactation on body composition and glucose metabolism in the mother-offspring pair. Both Reviewers find the data of interest. Whereas Reviewer 2 raises only relatively minor criticisms, Reviewer 1 expresses serious concerns regarding data interpretation and analysis. According to this Reviewer, the authors frequently overstate causality, particularly in linking maternal metabolic changes to fetal growth outcomes and hyperglycemia, when the data support only associations. Moreover, several aspects of the methods are judged to be incomplete or unclear, and statistical analysis is considered to be insufficient. These aspects should be convincingly addressed by the authors.

REFEREE COMMENTS

Referee #1:

This study addresses an important question, but substantial revisions are required. The manuscript currently overstates causality between maternal metabolism and fetal outcomes, and several key methodological details are missing. Statistical analyses need reworking, and figures should be clarified. Conclusions should be tempered, with greater consideration of placental nutrient transfer and existing literature. Overall, the dataset is valuable, but clarity, rigor, and balance in interpretation must be improved.

Major concerns:

- Maternal metabolism vs fetal growth: The manuscript does not clearly demonstrate whether altered maternal metabolism caused fetal growth issues, or whether these were simply associated. Alternative explanations, such as vascular or placental factors, should be considered.
- Fetal hyperglycemia: The authors attribute this to altered maternal metabolism, but it could also reflect nutrient transfer across the placenta, independent of maternal metabolic changes. Also, are the fetus really hyperglycemic? Normal glucose levels in mice?

- Causality: Current data do not support strong conclusions about maternal metabolic dysfunction directly driving fetal outcomes (DOHaD) - revisions should emphasize association rather than causation.

Additional comments:

Introduction

- The authors should explain how fetal nutrient supply was assessed and whether the study design adequately captured maternal-fetal nutrient dynamics.
- The statement "it is not known whether glucose intolerance is already present prior to pregnancy and worsens during pregnancy, or whether it persists during the lactational period" is not universally true across models. This section should be revised to acknowledge variation among models.

Methods

- GLU-Venus mice: Define this strain, justify its use, and explain how findings relate to other commonly used strains.
- HFHS diet: Introduced at 5 weeks-clarify how this timing relates to developmental milestones in mice.
- Fetal blood collection: The manuscript does not describe the method used-this must be included for reproducibility.
- Pup numbers and sex separation: Five pups per dam were analyzed, but the method of sex determination/separation is missing. This should be added.

Results & Figures

- Figures also lack clarity, with missing symbol identification and inadequate differentiation between maternal and offspring data.
- Figure 1:
 - o Current labeling is confusing. Statistical approach should be a two-way ANOVA with time (repeated measure) and diet as factors. Results should report main effects of time, diet, and their interaction, followed by appropriate post hoc testing. A statistical consultation is strongly recommended.
- Figures 2 & 3: Symbols are not identified in the legends - these must be clarified.
- Figure 6:
 - o Only three male fetuses showed higher glucose levels compared with controls, raising concerns about over-interpretation.
 - o Data presentation should differentiate maternal and offspring values more clearly, possibly using distinct symbols.

Discussion & Conclusion

- Line 610: The authors state maternal effects altered fetal growth, but results show they did not. This requires revision.
- Line 654: The phrase "these maternal effects resulted in fetal hyperglycemia" is too strong. Revise to reflect association, not causation.
- The maternal-fetal link is underdeveloped. Additional discussion of placental glucose transport mechanisms would strengthen the interpretation.
- Although extensive metabolic data were collected, the overall novelty of findings is limited. The discussion should be balanced, emphasizing incremental contributions rather than overstating novelty.
- Sex-specific outcomes are not discussed adequately.

Referee #2:

The authors attempt to evaluate the effect of a high-fat, high-sugar diet during pregnancy and lactation on body composition and glucose metabolism in the mother-offspring pair using a mouse model. The article is very well written and flows smoothly. Although the topic has been explored by other research groups, this study provides valuable information for further analysis and presents a novel approach. I have some recommendations that may be useful:

Results:

In the section "Greater energy expenditure in high-fat, high-sugar fed dams is accompanied by higher activity during the light but not the dark phase," you state: "During the light phase, both energy expenditure and activity were higher in HFHS dams compared to controls (Figure 3A, B)." In Figure 3B, however, there are no significant differences between the two groups in activity during the light phase, as you clearly mention in the following sentence. This inconsistency makes the section confusing. Additionally, you state: "During the dark phase, energy expenditure was higher in HFHS than control dams, with no differences in activity," which appears to be incorrect. According to the figure, activity during the dark phase was higher in the HFHS group at all stages.

In the section "High-fat, high-sugar feeding impairs glucose tolerance prior to pregnancy, which worsens during pregnancy and lactation," you mention that "the reduction in blood glucose concentration in response to an insulin challenge was comparable between the groups at GD 16 (Figure 5C)." However, in Figure 5C, the middle graph is labeled GD 18. Is this a labeling error?

You also state: "At the end of lactation, glucose tolerance remained impaired in HFHS... This was accompanied by higher plasma insulin concentrations in response to a glucose challenge (Figure 5A, B)." Yet, in the third graph of Figure 5B-the one corresponding to the weaning HFHS group-insulin levels appear to be lower.

Discussion:

Several paragraphs are overly long and contain multiple main ideas, which hinders the flow of the text.

END OF COMMENTS

To: Professor Bennet
Senior Editor, *The Journal of Physiology*

From: A/Prof Kathy Gatford
University of Adelaide, Australia

Re: Manuscript revision

Date: 1 December 2025

Dear Professor Bennet

Thank you for the opportunity to revise our manuscript “A high-fat, high-sugar diet impairs maternal metabolism throughout pregnancy and lactation in mice” by O’Hara, Gembus, Clarke, Page, Gatford and Nicholas, for publication in *The Journal of Physiology*. This manuscript is original, is not submitted for publication elsewhere, and has not been published on a pre-print server. All authors have read the revised manuscript and approve its submission and have no conflicts of interest to declare.

We thank the editor and reviewers for their valuable feedback. Each comment from the editor has been addressed below, with responses provided in italics below each original comment. In particular, we have added methodological information, clarified statistical analyses and revised figures as suggested by referee 1 and in response to comments from reviewer 2. Line numbers in curly brackets below refer to the tracked changes version of the revised manuscript.

We thank you for considering this revised article for publication in *The Journal of Physiology* and look forward to your response.

Sincerely,

A/Prof Kathryn Gatford on behalf of all co-authors

EDITOR COMMENTS

Reviewing Editor:

Comments to the Author:

The aim of the study is to evaluate, in a mouse model, the effect of a high-fat, high-sugar diet during pregnancy and lactation on body composition and glucose metabolism in the mother-offspring pair. Both Reviewers find the data of interest. Whereas Reviewer 2 raises only relatively minor criticisms, Reviewer 1 expresses serious concerns regarding data interpretation and analysis. According to this Reviewer, the authors frequently overstate causality, particularly in linking maternal metabolic changes to fetal growth outcomes and hyperglycemia, when the data support only associations. Moreover, several aspects of the methods are judged to be incomplete or unclear, and statistical analysis is considered to be insufficient. These aspects should be convincingly addressed by the authors.

Thank you for your comments and to the reviewers for their suggestions for improvement of the manuscript. Each point raised by the reviewers is addressed below.

REFEREE COMMENTS

Referee #1:

This study addresses an important question, but substantial revisions are required. The manuscript currently overstates causality between maternal metabolism and fetal outcomes, and several key methodological details are missing. Statistical analyses need reworking, and figures should be clarified. Conclusions should be tempered, with greater consideration of placental nutrient transfer and existing literature. Overall, the dataset is valuable, but clarity, rigor, and balance in interpretation must be improved.

Thank you for your careful review of the manuscript. We have addressed each comment directly below.

Major concerns:

- Maternal metabolism vs fetal growth: The manuscript does not clearly demonstrate whether altered maternal metabolism caused fetal growth issues, or whether these were simply associated. Alternative explanations, such as vascular or placental factors, should be considered.

Thank you for this comment. Interestingly, fetal weight and body size did not differ between dams fed control and HFHS diets, with the exception of biparietal diameter {Table 1}. Since head length was not different in fetuses from the two diet groups, this may have been a chance finding. Although placental weight and efficiency were not different between diets, we agree that measures of placental nutrient transport are altered in other mouse models of maternal obesity. Changes to placental function might therefore contribute to the higher glucose in HFHS fetuses compared to controls. Nevertheless, since glucose travels via facilitated transport down a concentration gradient from maternal to fetal circulations, and HFHS dams were glucose intolerant in late pregnancy, it seems logical to propose that the impaired maternal glucose tolerance might contribute to higher fetal glucose concentrations in late gestation. Potential contributions of both of these mechanisms are covered in the

discussion, and indeed this is indicated at the start of the paragraph, which does not attribute the cause of progeny changes as maternal metabolism; “Our measures in fetuses and neonatal offspring suggest that maternal HFHS-feeding disrupts fetal metabolism and early postnatal growth...”. In response to this comment, we have expanded the discussion of placental effects of HFHS diets. {discussion, lines 649-663}

- Fetal hyperglycemia: The authors attribute this to altered maternal metabolism, but it could also reflect nutrient transfer across the placenta, independent of maternal metabolic changes. Also, are the fetus really hyperglycemic? Normal glucose levels in mice?

Thank you for this suggestion. As noted above, we have revised and expanded the discussion of potential placental contributions to fetal glucose supply. Fetuses from HFHS-fed dams had higher glucose than those of control-diet-fed dams, as discussed further below. We have therefore described the HFHS fetuses as hyperglycaemic.

- Causality: Current data do not support strong conclusions about maternal metabolic dysfunction directly driving fetal outcomes (DOHaD) - revisions should emphasize association rather than causation.

We acknowledge placental as well as metabolic mechanisms may contribute to fetal and offspring outcomes. Potential roles of altered milk production and composition are also discussed as potential contributors to offspring outcomes. In response to this comment, we have revised the concluding paragraph to state association rather than causality between maternal effects and offspring outcomes. {discussion, lines 695-697}

Additional comments:

Introduction

- The authors should explain how fetal nutrient supply was assessed and whether the study design adequately captured maternal-fetal nutrient dynamics.

We acknowledge that we have only measured a single nutrient (glucose) in fetal blood and have revised the introductory statement summarising the outcomes studied in response to your comment. This now reads “We further characterised the model by assessing its impact on fetal circulating glucose and insulin concentrations and size and neonatal growth.” {lines 101-103}.

- The statement "it is not known whether glucose intolerance is already present prior to pregnancy and worsens during pregnancy, or whether it persists during the lactational period" is not universally true across models. This section should be revised to acknowledge variation among models.

This section has been revised and expanded as suggested to clarify the gaps in knowledge. {lines 83-96}

Methods

- GLU-Venus mice: Define this strain, justify its use, and explain how findings relate to other commonly used strains.

We have added this information as requested. {methods, lines 113-117}

- HFHS diet: Introduced at 5 weeks-clarify how this timing relates to developmental milestones in mice.

Maternal diets commenced at 4 weeks, approximately one week after weaning, when mice from the breeding colony are transferred to experimental holding areas. We have added this information to the methods as requested. {lines 124-126}

- Fetal blood collection: The manuscript does not describe the method used-this must be included for reproducibility.

Apologies for this omission – the information has been added to the methods {lines 146-148}.

- Pup numbers and sex separation: Five pups per dam were analyzed, but the method of sex determination/separation is missing. This should be added.

We have added information about sex determination in fetuses and neonates to the methods section as requested {lines 144-149 and 152-155}.

Results & Figures

- Figures also lack clarity, with missing symbol identification and inadequate differentiation between maternal and offspring data.

Please see responses below.

- Figure 1:

- o Current labeling is confusing.

We agree that the labelling becomes complicated where significant interactions between diet and day meant that posthoc testing was merited, or where a significant day effect meant that posthoc tests were needed to identify which days differed. Throughout all figures, we have used asterisks to indicate diet effects and letters to show differences between days. In

response to this comment and a suggestion from reviewer 2, we have made text reporting P-values larger and bolded where significant in all Figures to make the significant results easier to identify.

Statistical approach should be a two-way ANOVA with time (repeated measure) and diet as factors. Results should report main effects of time, diet, and their interaction, followed by appropriate post hoc testing. A statistical consultation is strongly recommended.

Thank you for this suggestion, which indeed is the statistical approach used to analyse data obtained in mice across multiple pregnancy timepoints, and separately for mice measured throughout lactation. Different groups of mice (with some overlap) are included in data presented for pre-mating, pregnancy and lactation data, because not all mice assessed before mating conceived and some mice were humanely killed at GD 18 pregnancy for fetal studies. We therefore analysed pre-mating, pregnancy and lactation data separately. In response to your comment, we have clarified that these three periods include different groups of mice within the figure legend {Figure 1 legend} and in the methods. {lines 169-171}.

- Figures 2 & 3: Symbols are not identified in the legends - these must be clarified.

The symbols used for control and HFHS mice are shown in a visual legend within each figure. In response to this suggestion we have also added the following sentence to each figure legend: "Data for control mice is shown in open symbols and data for HFHS mice is shown in filled symbols." {Figures & legends}

- Figure 6:

o Only three male fetuses showed higher glucose levels compared with controls, raising concerns about over-interpretation.

Thank you for this question. We acknowledge that there is variability within each group. The effect of maternal diet on fetal blood glucose concentrations did not differ between male and female offspring, such that fetal blood glucose was on average 34% higher in fetuses of HFHS-fed than control diet-fed dams. We have added this information to the results text. {page 30, lines 420-423}

o Data presentation should differentiate maternal and offspring values more clearly, possibly using distinct symbols.

Thank you for this suggestion. We have retained use of circles for maternal data and of the colour scheme throughout, consistent with our previous publications in this model (PMID: 41195747, PMID: 40023799). In response to this suggestion, we have now used triangles to show offspring data in the present manuscript. {Figure 6 and Figure 7}

Discussion & Conclusion

- Line 610: The authors state maternal effects altered fetal growth, but results show they did not. This requires revision.

Apologies – the sentence states early growth by which we meant neonatal growth. We agree that fetal growth was largely unaffected by maternal diet in the present study. In response to this comment, we have reworded this sentence to avoid confusion: “Our measures in fetuses and neonatal offspring suggest that maternal HFHS-feeding disrupts fetal metabolism and early postnatal growth”. {lines 633-635}

- Line 654: The phrase "these maternal effects resulted in fetal hyperglycemia" is too strong. Revise to reflect association, not causation.

We have reworded this sentence in the concluding paragraph as noted above “These maternal effects were associated with fetal hyperglycaemia, without hyperinsulinaemia and altered neonatal growth patterns of offspring, which were fatter at weaning”. {lines 695-697}

- The maternal-fetal link is underdeveloped. Additional discussion of placental glucose transport mechanisms would strengthen the interpretation.

We have expanded the discussion of placental structure and nutrient transport in HFHS mouse pregnancies as suggested. {discussion, lines 649-663}

- Although extensive metabolic data were collected, the overall novelty of findings is limited. The discussion should be balanced, emphasizing incremental contributions rather than overstating novelty.

The discussion extensively cites relevant prior literature to provide context for our findings, including the differential effects of shorter-term and less palatable diets with those that induce frank obesity. We identify the novel aspects of the present study throughout, e.g. the first assessment of substrate oxidation during mouse pregnancy.

- Sex-specific outcomes are not discussed adequately.

Interestingly, we did not observe any differential effects of HFHS exposure in male and female fetuses or neonates. We have added this point to the discussion, together with a summary of the available data on sex-specific impacts of maternal HFHS diet consumption on fetuses and neonates. {lines 633-645}

Referee #2:

The authors attempt to evaluate the effect of a high-fat, high-sugar diet during pregnancy and lactation on body composition and glucose metabolism in the mother-offspring pair

using a mouse model. The article is very well written and flows smoothly. Although the topic has been explored by other research groups, this study provides valuable information for further analysis and presents a novel approach. I have some recommendations that may be useful:

Results:

In the section "Greater energy expenditure in high-fat, high-sugar fed dams is accompanied by higher activity during the light but not the dark phase," you state: "During the light phase, both energy expenditure and activity were higher in HFHS dams compared to controls (Figure 3A, B)." In Figure 3B, however, there are no significant differences between the two groups in activity during the light phase, as you clearly mention in the following sentence. This inconsistency makes the section confusing.

The results text for this data is correct. Activity is higher in HFHS than control-diet-fed dams overall ($P = 0.007$, as shown in Figure 3B). We apologise that this is difficult to see and have revised all figures to increase font size of statistical comparisons. We have also bolded significant P -values to make these more visible.

Additionally, you state: "During the dark phase, energy expenditure was higher in HFHS than control dams, with no differences in activity," which appears to be incorrect. According to the figure, activity during the dark phase was higher in the HFHS group at all stages.

The results text for this data is correct. There is an overall diet effect for energy expenditure in the dark phase, which is higher in HFHS than control dams across pregnancy stages ($P < 0.001$, as shown in Figure 3A). There is no effect of diet on dam activity during the dark phase ($P = 0.149$, as shown in Figure 3B). Letters indicate differences between pregnancy stage; energy expenditure increases and activity decreases as pregnancy progresses.

In the section "High-fat, high-sugar feeding impairs glucose tolerance prior to pregnancy, which worsens during pregnancy and lactation," you mention that "the reduction in blood glucose concentration in response to an insulin challenge was comparable between the groups at GD 16 (Figure 5C)." However, in Figure 5C, the middle graph is labeled GD 18. Is this a labeling error?

Thank you for identifying the error. We have corrected the results text – ITT was performed at GD18, two days after GTT at GD16. {line 376}

You also state: "At the end of lactation, glucose tolerance remained impaired in HFHS... This was accompanied by higher plasma insulin concentrations in response to a glucose challenge (Figure 5A, B)." Yet, in the third graph of Figure 5B-the one corresponding to the weaning HFHS group-insulin levels appear to be lower.

The third (right-hand) panel of Figure 3B shows plasma insulin in dams during an intraperitoneal glucose challenge at weaning. Insulin concentrations are higher in the HFHS-

fed (closed symbols) than control diet-fed dams (open symbols), with a significant diet effect ($P = 0.011$, as shown in Figure 5B). It is only during pregnancy that we observed lower insulin concentrations in HFHS than control mice (middle panel), suggesting that the diet may impair pregnancy-induced upregulation of insulin secretion.

Discussion:

Several paragraphs are overly long and contain multiple main ideas, which hinders the flow of the text.

We apologise for the length of the discussion, which partly reflects the numerous outcomes we investigated and is also reflective of the in-text citation style. In response to your suggestion, we have broken several longer paragraphs into smaller sections.

Dear Dr Nicholas,

Re: JP-RP-2025-289985R1 "**A high-fat, high-sugar diet impairs maternal metabolism throughout pregnancy and lactation in mice**" by Stephanie Elise O'Hara, Kelly M Gembus, Georgia S Clarke, Amanda J Page, Kathryn L Gatford, and Lisa Marie Nicholas

Thank you for submitting your manuscript to The Journal of Physiology. It has been assessed by a Reviewing Editor and by 2 expert referees and we are pleased to tell you that it is acceptable for publication following satisfactory revision.

REVISION CHECKLIST:

Please upload two versions of your manuscript text: one with all relevant changes highlighted and one clean version with no changes tracked. The manuscript file should include all tables and figure legends, but each figure/graph should be uploaded as separate, high-resolution files. The journal is now integrated with Wiley's Image Checking service. For further details, see: <https://www.wiley.com/en-us/network/publishing/research-publishing/trending-stories/upholding-image-integrity-wileys->

image-screening-service

We look forward to receiving your revised submission.

Yours sincerely,

Laura Bennet
Senior Editor
The Journal of Physiology

EDITOR COMMENTS

Reviewing Editor:

Comments to the Author:

Whereas Reviewer 1 is satisfied with the revision, Reviewer 2 points to some inconsistencies in data presentation which should be convincingly addressed by the authors.

REFEREE COMMENTS

Referee #1:

The authors have adequately addressed my previous comments.

Referee #2:

Thank you to the authors for addressing my previous comments. However, several sections of the manuscript remain unclear and potentially contradictory.

First, the authors state that "there is no effect of diet on dam activity during the dark phase ($P = 0.149$, as shown in Figure 3B)." However, in Figure 3B, during pregnancy and specifically in the dark phase, the HFHS group exhibits the highest mean activity levels in both pre-pregnancy and early pregnancy. Moreover, the figure indicates a significant stage effect ($p < 0.01$), which appears to suggest increased activity during the dark phase under the HFHS diet in these conditions. If this interpretation is not correct, the figure is difficult to understand in its current form, and I would recommend restructuring or clarifying the figure and its legend. As presented, the statement that there is no greater activity associated with diet is confusing and seems inconsistent with the data shown. Further clarification or a more precise explanation of these results is needed.

Second, Figure 5B consists of four panels. The first three panels display insulin response curves following an intraperitoneal glucose challenge at different time points. In panel 3 (weaning), the HFHS group shows higher insulin levels compared to the control group. However, in panel 4, which summarizes the area under the curve (AUC), the HFHS group exhibits a lower AUC at weaning. This appears inconsistent with the insulin response observed in panel 3. In addition, the reported p-value for this comparison is 0.130, indicating that the difference is not statistically significant. Consequently, the relationship between the insulin curves and the AUC results at weaning is unclear, and these findings do not appear to be fully consistent. The authors should clarify how these results should be interpreted.

END OF COMMENTS

To: Professor Bennet
Senior Editor, *The Journal of Physiology*

From: A/Prof Kathy Gatford
University of Adelaide, Australia

Re: Manuscript revision

Date: 5 January 2026

Dear Professor Bennet

Thank you for the opportunity to further revise our manuscript “A high-fat, high-sugar diet impairs maternal metabolism throughout pregnancy and lactation in mice” by O’Hara, Gembus, Clarke, Page, Gatford and Nicholas, for publication in *The Journal of Physiology*. This manuscript is original, is not submitted for publication elsewhere, and has not been published on a pre-print server. All authors have read the revised manuscript and approve its submission and have no conflicts of interest to declare.

We thank the editor and reviewers for their valuable feedback. The two remaining points raised by reviewer 2 have been addressed below, with responses provided in italics below each original comment. Line numbers in curly brackets below refer to the tracked changes version of the revised manuscript. In particular, we have clarified the description of results for maternal activity in Figure 3 and clarified analyses and provided additional interpretation of effects of diet on circulating insulin concentrations vs response to glucose challenge in Figure 5.

We thank you for considering this revised article for publication in *The Journal of Physiology* and look forward to your response.

Sincerely,

A/Prof Kathryn Gatford on behalf of all co-authors

EDITOR COMMENTS

Reviewing Editor:

Comments to the Author:

Whereas Reviewer 1 is satisfied with the revision, Reviewer 2 points to some inconsistencies in data presentation which should be convincingly addressed by the authors.

Thank you for your comments and to the reviewers for their suggestions for improvement of the manuscript. The remaining points raised by reviewer 2 are addressed below.

REFEREE COMMENTS

Referee #1:

The authors have adequately addressed my previous comments.

Thank you for your suggestions to improve the previous version of the manuscript.

Referee #2:

Thank you to the authors for addressing my previous comments. However, several sections of the manuscript remain unclear and potentially contradictory.

Thank you for your suggestions to improve the previous version of the manuscript. The points raised in relation to the R1 manuscript are addressed below.

First, the authors state that "there is no effect of diet on dam activity during the dark phase ($P = 0.149$, as shown in Figure 3B)." However, in Figure 3B, during pregnancy and specifically in the dark phase, the HFHS group exhibits the highest mean activity levels in both pre-pregnancy and early pregnancy. Moreover, the figure indicates a significant stage effect ($p < 0.01$), which appears to suggest increased activity during the dark phase under the HFHS diet in these conditions. If this interpretation is not correct, the figure is difficult to understand in its current form, and I would recommend restructuring or clarifying the figure and its legend. As presented, the statement that there is no greater activity associated with diet is confusing and seems inconsistent with the data shown. Further clarification or a more precise explanation of these results is needed.

Thank you for your feedback on this figure (copied below for reference, including legend). We respectfully disagree with this feedback that "the statement that there is no greater activity associated with diet is confusing and seems inconsistent with the data shown". As shown in Figure 3B, activity during the dark phase did not differ between diet groups. As there is no significant interaction between diet and stage, performing post-hoc tests to compare diet groups within each pregnancy stage would be statistically inappropriate. As

can be seen in the Figure 3B dark phase panel, individual data points for Control and HFHS groups overlap substantially at all pregnancy stages, which is consistent with the lack of a difference between the diet groups. Although dark-phase activity did not differ between diet groups, there was a significant effect of pregnancy stage ($P < 0.001$) for activity during the dark phase.

In response to your comment, we have expanded the text describing these results, to hopefully reduce confusion in the description of diet and stage effects. This now reads:

“During the light phase, both energy expenditure and activity were higher in HFHS dams compared to controls (Figure 3A, B). Light phase energy expenditure steadily increased as pregnancy progressed, whilst activity remained stable before and during pregnancy. During the dark phase, energy expenditure was higher in HFHS than control dams, without differences in activity between diet groups (Figure 3A, B). Dark phase energy expenditure differed between pregnancy stages, increasing from mid-pregnancy onwards (Figure 3A). Activity during the dark phase also varied across pregnancy stages (Figure 3B), falling by ~50% at early- and mid-pregnancy relative to pre-mating activity and decreasing further at late pregnancy.” {lines 308-317}

Figure 3: Energy expenditure (A) and activity (B) during a 12-hour light and 12-hour dark phase in mice fed a control or high-fat, high-sugar (HFHS) diet for 11 weeks before mating and throughout pregnancy and lactation. Data were collected for four days prior to mating and from mating until gestational day 17.5 ($n = 9$ Control, 8 HFHS mice) and were analysed by 2-way repeated measures ANOVA; symbols show individual values with lines and whiskers showing means and standard deviation for each group. Data for control mice is shown in open symbols and data for HFHS mice is shown in filled symbols. Where stage effects were

significant, Bonferroni comparison was used to compare pairs of stages. Differences between stages are indicated with letters (overall: j, k, l, m).

Second, Figure 5B consists of four panels. The first three panels display insulin response curves following an intraperitoneal glucose challenge at different time points. In panel 3 (weaning), the HFHS group shows higher insulin levels compared to the control group. However, in panel 4, which summarizes the area under the curve (AUC), the HFHS group exhibits a lower AUC at weaning. This appears inconsistent with the insulin response observed in panel 3. In addition, the reported p-value for this comparison is 0.130, indicating that the difference is not statistically significant. Consequently, the relationship between the insulin curves and the AUC results at weaning is unclear, and these findings do not appear to be fully consistent. The authors should clarify how these results should be interpreted.

Thank you for this comment, which likely arises due to inconsistencies amongst the field in how area under the curve is calculated. We have followed best practice, calculating the area relative to fasting concentrations, reflective of the response to glucose. The lack of a diet effect on insulin AUC therefore reflects higher fasting insulin concentrations, subtracted from higher insulin concentrations during the response phase. In the right-hand panel of Figure 5B, the analysis shown is for effects of diet, stage and interaction – the noted P-value of 0.130 for diet is across all stages, while insulin AUC also changed between stages.

In response to your comment, we have clarified the description of these results to more clearly describe insulin concentrations as being higher before and during the glucose challenge {lines 379-380} and to include a brief description of the approach used to determine insulin response “assessed as insulin profile area above fasting insulin” {lines 385-386}.

We have also added information about these calculations to the methods section describing the ipGTT and ipITT:

“Areas under the curve for glucose and insulin responses to glucose challenge were calculated relative to fasting concentrations of glucose and insulin, measured directly before the intraperitoneal glucose tolerance test. Area above the curve for the glucose response to insulin challenge was similarly calculated relative to fasting glucose measured directly before the intraperitoneal insulin tolerance test.” {lines 206-210}

We have added a sentence to the discussion interpreting this result as requested:

“In contrast, although we did not see evidence of an enhanced insulin response to a glucose challenge, the higher insulin concentrations in HFHS mice before and throughout glucose challenge suggests increased insulin secretion overall.” {lines 593-596}

Dear Dr Nicholas,

Re: JP-RP-2026-289985R2 "**A high-fat, high-sugar diet impairs maternal metabolism throughout pregnancy and lactation in mice**" by Stephanie Elise O'Hara, Kelly M Gembus, Georgia S Clarke, Amanda J Page, Kathryn L Gatford, and Lisa Marie Nicholas

We are pleased to tell you that your paper has been accepted for publication in The Journal of Physiology.

Yours sincerely,

Laura Bennet
Senior Editor
The Journal of Physiology

IMPORTANT POINTS TO NOTE FOLLOWING ACCEPTANCE OF YOUR PAPER:

- **IMPORTANT NOTICE ABOUT OPEN ACCESS:** To assist authors whose funding agencies mandate immediate public access to published research findings, The Journal of Physiology allows authors to pay an Open Access (OA) fee to have their papers made freely available immediately on publication.

- You can help your research get the attention it deserves! Check out Wiley's free Promotion Guide for best-practice recommendations for promoting your work at: www.wileyauthors.com/eoo/guide. You can learn more about Wiley Editing Services which offers professional video, design, and writing services to create shareable video abstracts, infographics, conference posters, lay summaries, and research news stories for your research at: www.wileyauthors.com/eoo/promotion.

- If you would like to receive our 'Research Roundup', a monthly newsletter highlighting the cutting-edge research published in The Physiological Society's family of journals (The Journal of Physiology, Experimental Physiology, Physiological Reports, The Journal of Nutritional Physiology and The Journal of Precision Medicine: Health and Disease), please click this link, fill in your name and email address and select 'Research Roundup': <https://www.physoc.org/journals-and-media/membernews>

EDITOR COMMENTS

Reviewing Editor:

Comments to the Author:

The referees are satisfied with the revision. Congratulations for an interesting study.

REFEREE COMMENTS

Referee #2:

The authors have adequately answered my questions about the interpretation of the figures.